# A predominant enhancer co-amplified with the *SOX2* oncogene is necessary and sufficient for its expression in squamous cancer

Yanli Liu[1,2,3,14], Zhong Wu[4,5,14], Jin Zhou[4,5,14], Dinesh K. A. Ramadurai[2], Katelyn L. Mortenson [2], Estrella Aguilera-Jimenez [2], Yifei Yan[6,7], Xiaojun Yang[3], Alison M. Taylor[8], Katherine E. Varley[2], Jason Gertz[2], Peter S. Choi[9], Andrew D. Cherniack [5,10], Xingdong Chen[1,11,12], Adam J. Bass [5,10,13], Swneke D. Bailey [6,7✉] & Xiaoyang Zhang [1,2✉]

Amplification and overexpression of the *SOX2* oncogene represent a hallmark of squamous cancers originating from diverse tissue types. Here, we find that squamous cancers selectively amplify a 3′ noncoding region together with *SOX2*, which harbors squamous cancer-specific chromatin accessible regions. We identify a single enhancer e1 that predominantly drives *SOX2* expression. Repression of e1 in *SOX2*-high cells causes collapse of the surrounding enhancers, remarkable reduction in *SOX2* expression, and a global transcriptional change reminiscent of *SOX2* knockout. The e1 enhancer is driven by a combination of transcription factors including SOX2 itself and the AP-1 complex, which facilitates recruitment of the co-activator BRD4. CRISPR-mediated activation of e1 in *SOX2*-low cells is sufficient to rebuild the e1-*SOX2* loop and activate *SOX2* expression. Our study shows that squamous cancers selectively amplify a predominant enhancer to drive *SOX2* overexpression, uncovering functional links among enhancer activation, chromatin looping, and lineage-specific copy number amplifications of oncogenes.

[1] State Key Laboratory of Genetic Engineering, School of Life Sciences, Fudan University, Shanghai, China. [2] Department of Oncological Sciences, Huntsman Cancer Institute, University of Utah, Salt Lake City, UT, USA. [3] College of Animal Science and Technology, Northwest Agriculture and Forestry University, Yangling, Shanxi, China. [4] Department of Surgery, West China Hospital, Sichuan University, Chengdu, Sichuan, China. [5] Department of Medical Oncology, Dana-Farber Cancer Institute, Harvard Medical School, Boston, MA, USA. [6] Cancer Research Program, Research Institute of the McGill University Health Centre, Montreal, QC, Canada. [7] Departments of Surgery and Human Genetics, McGill University, Montreal, QC, Canada. [8] Department of Pathology and Cell Biology, Herbert Irving Comprehensive Cancer Center, Columbia University, New York, NY, USA. [9] Department of Pathology and Laboratory Medicine, Perelman School of Medicine, University of Pennsylvania, Philadelphia, PA, USA. [10] Broad Institute of MIT and Harvard, Cambridge, MA, USA. [11] Fudan University Taizhou Institute of Health Sciences, Taizhou, Jiangsu, China. [12] Yiwu Research Institute of Fudan University, Yiwu, Zhejiang, China. [13] Department of Medicine, Herbert Irving Comprehensive Cancer Center, Columbia University, New York, NY, USA. [14] These authors contributed equally: Yanli Liu, Zhong Wu, Jin Zhou. ✉email: swneke.bailey@mcgill.ca; xiaoyang_zhang@fudan.edu.cn

Lineage-specific oncogenes represent a class of genes that play important roles in the normal development of specific cell lineages, but drive tumorigenesis when dysregulated. Many lineage-specific oncogenes encode transcription factors such as MITF in melanomas[1], AR in prostate cancer[2], CDX2 in colorectal cancer[3], and KLF5 in squamous cancer and colorectal cancer[4,5]. SOX2, a member of the SRY-box transcription factor family, is well known for its role in the pluripotency of embryonic stem cells (ESCs)[6]. SOX2 is also essential in maintaining the self-renewal ability of basal cells[7], which have been reported as the cell of origin for squamous cancers, the most common type of solid tumors[8]. Squamous cancer can originate from diverse tissues such as lung (lung squamous cell carcinoma; LUSC), cervix (cervical squamous cell carcinoma; CESC), skin, esophagus (esophageal squamous cell carcinoma; ESSC), and upper digestive tissues in the head and neck (head and neck squamous cell carcinoma; HNSC). Genomic analyses have revealed that the SOX2 gene is widely amplified and overexpressed in squamous cancers, nominating SOX2 as a lineage-specific oncogene[9–16]. Indeed, previous in vivo studies have shown that Sox2 overexpression, together with the inactivation of tumor suppressors such as Pten or Lkb1, drives the formation of mouse lung squamous cancers[8,17,18]. In addition to squamous cancer, SOX2 amplification and overexpression have also been reported in glioma[19], a common type of brain tumor that includes low-grade glioma (LGG), glioblastoma (GBM), and several other subtypes.

SOX2 overexpression in cancer cells has been largely attributed to copy number amplifications of the SOX2 gene itself[9–11,14–16,19]. However, our understanding of oncogene copy number amplifications is evolving. We and others have recently shown that noncoding enhancers outside oncogenes such as MYC, MYCN, AR, KLF5, and EGFR are selectively amplified with or without their respective oncogenes[5,20–28], demonstrating novel mechanisms that transcriptionally activate oncogenes in diverse cancer types. Therefore, we decided to revisit the SOX2 locus and its associated copy number changes.

Here, we reveal distinct copy number profiles at the SOX2 locus between squamous cancers and gliomas, which corresponds to the distribution of lineage-specific potential regulatory elements. Focusing on the noncoding region that is selectively co-amplified with SOX2 in squamous cancers, we discover a single predominant enhancer that is necessary and sufficient for SOX2 activation. Furthermore, we delineate its relationship with the surrounding enhancers, identify its associated transcription factors, reveal its vulnerability to bromodomain protein degradation, and illustrate its impact on 3D chromatin architecture. Our study reveals the functional link among enhancer activation, enhancer–promoter interactions, and lineage-specific copy number amplifications in cancer.

## Results

**Squamous cancers selectively amplify lineage-specific chromatin-accessible noncoding regions adjacent to the SOX2 oncogene.** To identify regions that are recurrently amplified in squamous cancers, we applied Genomic Identification of Significant Targets in Cancer (GISTIC)[29] to the single-nucleotide polymorphism (SNP)-array-based copy number data of combined squamous cancer samples (LUSC, ESSC, HNSC, and CESC) from The Cancer Genome Atlas (TCGA)[11,30,31] (Supplementary Fig. 1a). In comparison, we also analyzed copy number data of gliomas (combined LGG and GBM), another cancer type that is associated with SOX2 overexpression. The GISTIC focal amplification peaks showed that, although both of the cancer types significantly amplify the SOX2 gene, they also selectively amplify noncoding regions adjacent to SOX2. Indeed,

the squamous cancer peak (chr3:181,415,947–181,719,852) covers SOX2 and a ~290 kb noncoding region 3′ to SOX2, while the glioma peak (chr3:181,256,575–181,496,100) covers SOX2 and a ~173 kb noncoding region 5′ to SOX2 (Fig. 1a). Squamous cancers are known to acquire arm-level or broad amplifications at the chromosome 3q arm where SOX2 resides[32]. We selected focal copy number alterations that are smaller than 10 Mb. Focusing on samples that harbor focal amplifications of SOX2 (amplitude log 2 (copy number/2) > 0.1), which corresponds to 9% of squamous cancers and 4% of gliomas, we profiled the averaged copy number amplitude surrounding the SOX2 locus (Fig. 1a and Supplementary Fig. 1b). The copy number profiles agree with the GISTIC results, showing cancer type-specific amplifications of the noncoding regions 3′ and 5′ to SOX2 (Fig. 1a and Supplementary Fig. 1c). TCGA squamous cancers and gliomas with SOX2 focal amplifications are associated with higher SOX2 expression, as compared to samples with non-focal amplifications or samples without amplifications (Supplementary Fig. 2). SOX2 overlaps with the SOX2-OT noncoding gene (Fig. 1b). We found that SOX2 focally amplified squamous cancers are also associated with higher SOX2-OT expression, which was not observed in gliomas (Supplementary Fig. 2).

We hypothesized that the distinct copy number profiles between these cancers may be attributed to lineage-specific distribution of regulatory elements. Therefore, we analyzed the assay of transposase accessible chromatin-sequencing (ATAC-seq) data from TCGA, which profiled genome-wide chromatin accessibility to identify potential regulatory elements in diverse types of human primary tumors[33]. In squamous cancers, we found multiple chromatin-accessible regions within the 3′ noncoding region that is selectively amplified in squamous cancers (Fig. 1b). These regions exhibit little chromatin accessibility in gliomas, suggestive of their squamous cancer-specific function (Fig. 1b). In contrast, most of the glioma-specific chromatin-accessible regions, as defined by the ATAC-seq signal, are distributed in the 5′ noncoding region that is selectively amplified in gliomas (Fig. 1b). We then calculated cancer specificity Z-scores for each of the TCGA-annotated chromatin-accessible sites by comparing the ATAC-seq signal across all the profiled cancer types, which highlighted the unique spatial distribution of squamous cancer- and glioma-specific chromatin accessibility (Fig. 1c, examples of highlighted regions are shown in Fig. 1d). Collectively, these data suggest that these two cancer types may selectively amplify lineage-specific regulatory elements together with the SOX2 oncogene (Fig. 1e).

**3D genomics identified SOX2 candidate functional enhancers in squamous cancer cells.** We next sought to interrogate the relationship between the SOX2 gene and the potential regulatory elements. The human genome is organized into series of insulated neighborhoods or topologically associating domains demarcated by CCCTC-binding factor (CTCF) binding, which restrict promoter–enhancer interactions[34]. We found that the SOX2 promoter resides at the boundary of two adjacent insulated neighborhoods that were previously identified from CTCF chromatin interaction analysis with paired-end tag (ChIA-PET) sequencing analysis[35], suggesting that the SOX2 promoter may have access to regulatory elements from both ends (Fig. 2a). Indeed, we observed strong CTCF binding sites as well as DNA motifs of other chromatin looping factors such as YY1[36] and ZNF143[37] in front of the SOX2 promoter region (Fig. 2a and Supplementary Fig. 3a), suggesting that the SOX2 promoter region may serve as a docking site for chromatin loops.

Given the high frequency of SOX2 focal amplifications in squamous cancers, in addition to previous in vivo evidence

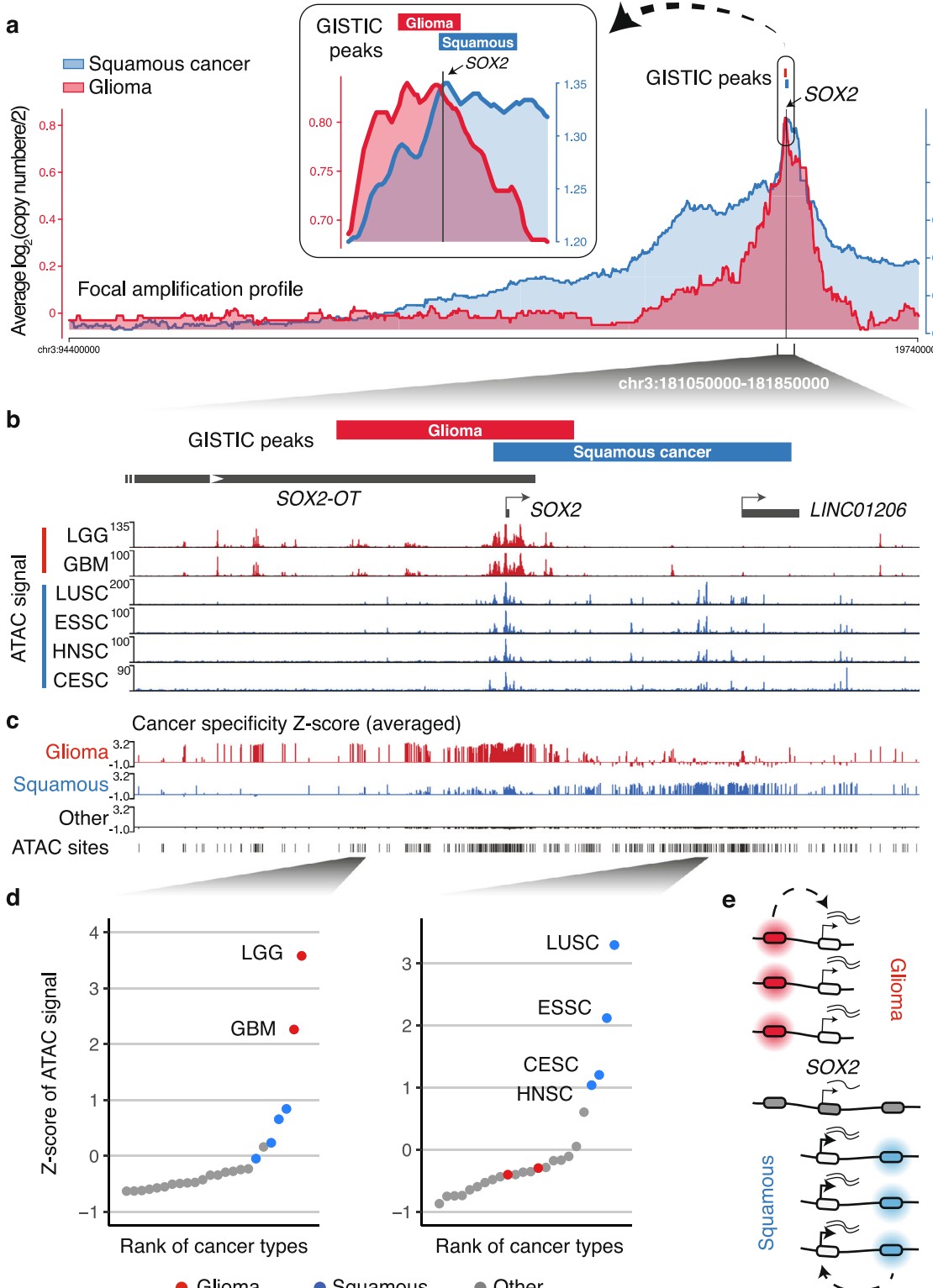

**Fig. 1 Copy number amplifications of lineage-specific enhancers 5′ and 3′ to the *SOX2* gene. a** GISTIC peaks for glioma and squamous cancers are presented as red and blue bars, respectively. Presented underneath the GISTIC peaks are the profiles of the averaged copy number score (log 2 (copy number/2)) for focal amplifications (<10 Mb) from samples that are associated with *SOX2* amplifications (log 2 (copy number/2) > 0.1). **b** TCGA normalized ATAC-seq profile— averaged accessibility per 100 bp-bin across multiple samples in LGG, GBM, LUSC, ESSC, HNSC, and CESC samples. **c** For each of the TCGA-annotated chromatin-accessible sites, *Z*-scores of ATAC-seq signal were calculated across all the TCGA-profiled cancer types. The averaged *Z*-scores for gliomas (GBM and LGG), squamous cancers (LUSC, ESSC, HNSC, and CESC), and other cancer types are presented. **d** The *Z*-scores of ATAC-seq signal for highlighted chromatin-accessible regions across the TCGA-profiled cancer types. **e** Schematic illustrating that gliomas and squamous cancers may selectively amplify lineage-specific regulatory elements together with the *SOX2* oncogene. Source data are provided as a Source Data file.

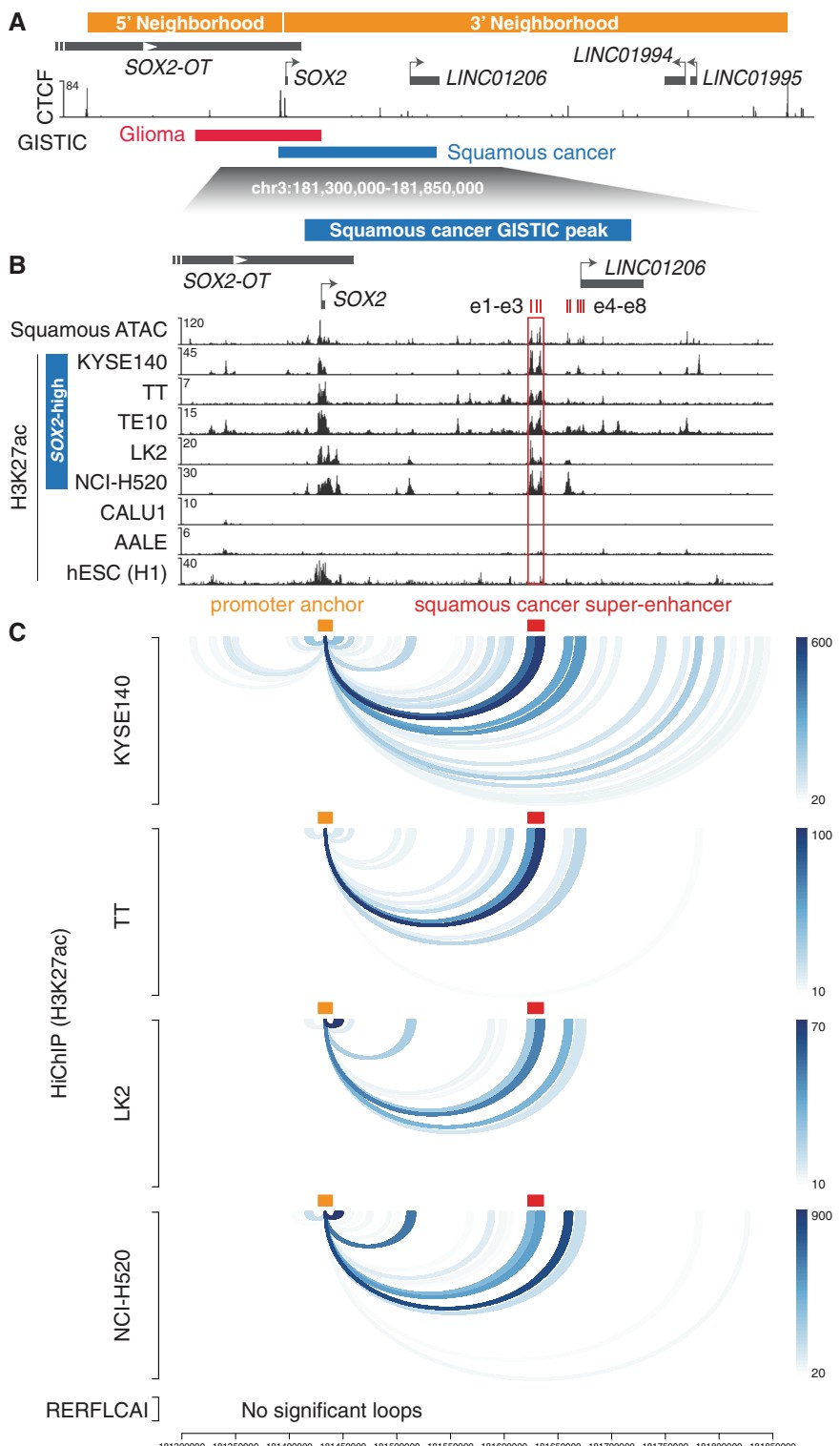

**Fig. 2 Identification of *SOX2* candidate functional enhancers in squamous cancer cells. a** CTCF ChIP-seq signal—signal per million reads (SPMRs) per bp in KYSE140 cells are presented underneath the positions of insulated neighborhoods derived from CTCF ChIA-PET data[35]. **b** Averaged ATAC-seq signal from TCGA squamous cancers and H3K27ac ChIP-seq signal (SPMR/bp) from squamous cancer cell lines, AALE and hESC cells are presented at the highlighted *SOX2* locus. The super-enhancer region shared by the listed *SOX2*-high squamous cancer cell lines is highlighted. **c** H3K27ac HiChIP loops that are connected to the *SOX2* promoter in squamous cancer cell lines. HiChIP anchors are defined by the union of H3K27ac peaks identified from the *SOX2*-high squamous cancer cell lines listed in (**b**). The color intensity corresponds to the number of PETs supporting each of the loops.

demonstrating the oncogenic significance of *SOX2* overexpression in squamous cancer[8,17,18], we focused on characterizing the functional importance of the squamous cancer-specific chromatin-accessible regions co-amplified with *SOX2*. We first aimed to assess their enhancer activity and physical interaction with the *SOX2* gene promoter. We analyzed chromatin immunoprecipitation-sequencing (ChIP-seq) data of H3K27ac, a marker for active regulatory elements, in esophageal squamous cancer cell lines KYSE140, TT, and TE10, and lung squamous cancer cell lines LK2 and NCI-H520[5,38] (Fig. 2b). All of these cell lines are associated with high *SOX2* expression and co-amplification of *SOX2* and the candidate enhancers (Supplementary Fig. 3b). We found that several of the squamous cancer-specific chromatin-accessible regions exhibit strong and consistent H3K27ac signals (Fig. 2b). In particular, three individual candidate enhancers, which we refer to as e1–e3, form a super-enhancer element (chr3:181,624,870–181,635,218) as defined by strong and condensed enrichment of H3K27ac signal across *SOX2*-high squamous cancer cell lines (Fig. 2b). Nearby candidate enhancers e4–e5 are also enriched with varying levels of H3K27ac in these cell lines (Fig. 2b), while e6–e8 show noticeable H3K27ac signal only in KYSE140 and TE10 cells. The e6–e8 elements reside within ±5 kb of the transcription start site (TSS) of *LINC01206*, suggesting that they may serve as the promoter or promoter-proximal elements of the noncoding gene (Fig. 2a).

In contrast, little H3K27ac signal was detected at these regions in the LUSC cell line CALU1, which exhibits low *SOX2* expression, or the immortalized normal lung epithelial cell line AALE[39] (Fig. 2b and Supplementary Fig. 3b). Interestingly, human ESCs that are associated with high *SOX2* expression also exhibit little H3K27ac signal at these loci[40] (Fig. 2b). Previous studies have identified a super-enhancer element that drives *Sox2* expression in mouse ESC[41,42]. We applied LiftOver[43] to identify mouse genomic regions that are conserved to the human squamous cancer-specific enhancers including the e1–e3 super-enhancer. We found that they are distinct from the reported mouse ESC super-enhancer (Supplementary Fig. 3c). Our findings suggest that this set of candidate enhancers is specific to *SOX2*-high squamous cancer cells.

We then applied H3K27ac HiChIP assays[44] to assess the physical interactions (false discovery rate (FDR) < 0.05) of the candidate enhancers with the *SOX2* promoter in *SOX2*-high esophageal squamous cancer cell lines KYSE140, KYSE70, and TT and lung squamous cancer cell lines LK2 and NCI-H520 (Fig. 2c and Supplementary Fig. 3d). The results consistently show that among the enhancers, the super-enhancer constituents e1–e3 have the strongest physical interaction with the *SOX2* promoter region. In contrast, these interactions are absent in the LUSC cell line RERFLCAI that exhibits low *SOX2* expression (Fig. 2c). Taken together, these data support e1–e3 as likely functional enhancers of the *SOX2* oncogene in squamous cancers.

**SOX2 activation is predominantly driven by a single enhancer in squamous cancer.** Focusing on the enhancers e1–e3 within the squamous cancer-specific super-enhancer as well as the adjacent enhancers e4–e5, we sought to interrogate their impact on *SOX2* expression. We applied an improved CRISPR interference (CRISPRi) system, which uses an inactivated Cas9 (dCas9) fused to two transcriptional repressors KRAB and MeCP2[45], to inhibit each of the five enhancers in *SOX2*-high squamous cancer cell lines KYSE140, LK2, and NCI-H520. ChIP-coupled with quantitative PCR (ChIP-qPCR) of dCas9 in KYSE140 cells validated the on-target effects of the single guide RNAs (sgRNAs) (Supplementary Fig. 4a). We found that repression of the e1 enhancer, but not the other four enhancers, consistently resulted in remarkable reductions (64–75%) in *SOX2* expression, suggesting

the predominant role of e1 (Fig. 3a). For validation, we included a separate sgRNA to target e1 and performed CRISPRi assays in six *SOX2*-high squamous cancer cell lines representing three tissue types ESSC, LUSC, and HNSC. We showed that e1 repression consistently led to 62–82% reductions in *SOX2* expression (averaged value of two separate sgRNAs) across the six cell lines (Fig. 3b). In addition, repression of e1 resulted in clear reductions in the protein levels of SOX2, as revealed by immunoblotting, in all the six cell lines (Fig. 3c). Previous studies have shown that the proliferation of squamous cancer cells with *SOX2* overexpression are dependent on the *SOX2* gene[9,46]. We showed that e1 repression led to significant reductions in the cell proliferation rate of *SOX2*-high squamous cancer cell lines KYSE140, LK2, and NCI-H520 (Fig. 3d). The proliferation-inhibitory phenotype observed in KYSE140 cells was rescued by ectopically expressing *SOX2* (Supplementary Fig. 4b), indicating that e1 regulates cell proliferation through activating *SOX2*. We also transplanted the LK2 cells with and without e1 repression into flanks of nude mice, which showed that activity of the e1 enhancer is required for in vivo tumor growth (Fig. 3e).

The *SOX2* gene encodes an SRY-box transcription factor that is involved in both transcriptional activation and repression[47]. We thereby reasoned that, in addition to reduced *SOX2* expression, repression of e1 may also result in dysregulation of SOX2-associated gene expression programs. We first identified SOX2-activated and -repressed genes by performing RNA-seq assays in the ESSC KYSE140 cells with and without CRISPR-mediated *SOX2* knockouts (Supplementary Fig. 4c). We selected the top 1000 genes that are activated or repressed by SOX2 (FDR-ranked; FDR < 0.05) and performed Gene Set Enrichment Analysis (GSEA), which showed that e1 repression caused expression changes of these genes in a manner that is highly similar to that caused by *SOX2* knockouts (Fig. 3f). Indeed, e1 repression significantly downregulated expression of SOX2-activated genes (normalized enrichment score = −2.06, P < 0.001) and upregulated SOX2-repressed genes (normalized enrichment score = 1.85, P < 0.001). Furthermore, expression of e1-regulated genes (FDR < 0.05; fold change > 1.5) is significantly correlated with *SOX2* expression in TCGA squamous cancer samples, suggesting they are likely to be SOX2-target genes in human primary tumors (Fig. 3g). Genes activated by e1 are enriched in squamous cancer-related pathways such as MAPK signaling and Hedgehog signaling (Supplementary Table 1). Collectively, these results demonstrate the critical role of the e1 enhancer in *SOX2* activation and SOX2-associated cellular and molecular phenotypes.

We then went on to test if e1 and the surrounding enhancers directly regulate any other genes in addition to *SOX2*. We analyzed the HiChIP data in *SOX2*-high squamous cancer cell lines by focusing on HiChIP anchors that harbor the e1–e8 elements (four anchors in total). We identified four additional candidate coding and noncoding genes *FXR1*, *ATP11B*, *SOX2-OT*, and *LINC01206*—the promoter region of each gene interacts with at least one of the enhancer anchors in two or more of the five tested cell lines (Supplementary Fig. 5a). Among them, the *SOX2* promoter has the strongest interactions with these enhancer anchors. We then performed CRISPRi assays in KYSE140 to assess the effects of e1–e8 on these candidate genes. In addition to *SOX2*, e1 repression also decreased *SOX2-OT* expression (Supplementary Fig. 5b). However, ectopic expression of *SOX2*, which had no effect on the decreased endogenous *SOX2* expression, rescued the decreased *SOX2-OT* expression (Supplementary Fig. 5c). This result, together with our observation of several SOX2 binding sites at or next to *SOX2-OT* promoter region (Supplementary Fig. 5d), suggests that *SOX2-OT* is directly regulated by SOX2 but not e1. Repression of e6–e8 caused

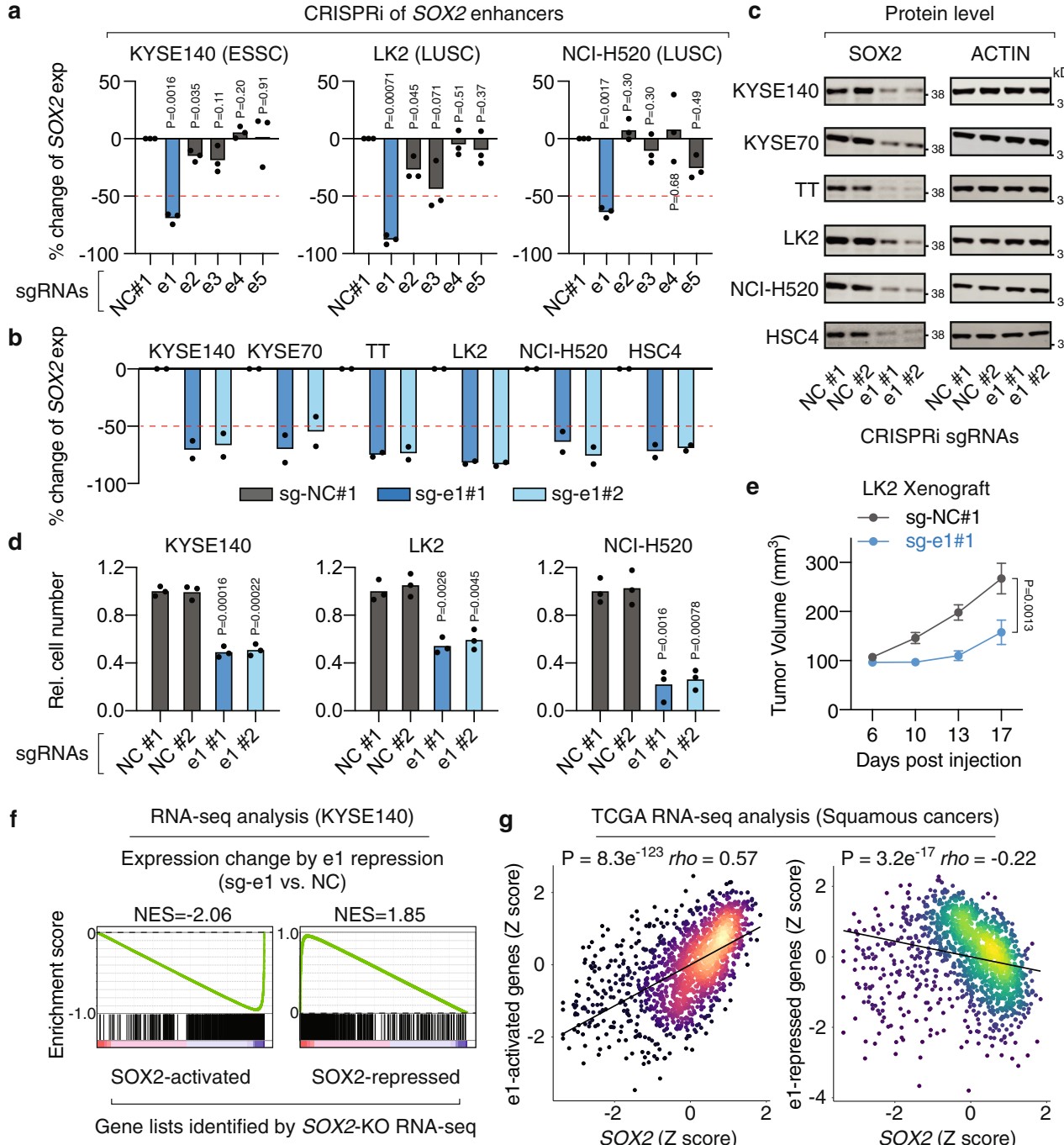

**Fig. 3 Identification of a predominant single enhancer driving *SOX2* overexpression. a** RT-qPCR measuring expression changes of *SOX2* after CRISPR-mediated repression of each of the enhancers e1–e5. The expression level of *SOX2* was normalized to cells treated with sg-NC#1 that has no recognition site in the genome. $n = 3$ biologically independent experiments. *P* values are derived from two-sided *t* tests. **b** RT-qPCR measuring expression changes of *SOX2* after CRISPR-mediated repression of the e1 enhancer. Two separate sgRNAs sg-e1#1 and sg-e1#2 were used to target the e1 enhancer in six squamous cancer cell lines. $n = 2$ biologically independent experiments for each sgRNA. **c** Immunoblots showing protein level changes of SOX2 after repression of the e1 enhancer in six squamous cancer cell lines. The ACTIN protein serves as a loading control. The immunoblotting experiment was repeated once independently with similar results. **d** Cell proliferation of squamous cancer cell lines with and without e1 repression. Cell numbers were counted 6 days post seeding and normalized to the sg-NC#1 control. $n = 3$ biologically independent experiments. *P* values are derived from two-sided *t* tests. **e** In vivo tumor growth of xenografts that were derived from LK2 cells with and without e1 repression. Error bar: standard error of the mean. $n = 10$ biologically independent samples of each condition. *P* value was derived from a two-way ANOVA with repeated measures. **f** GSEA analysis measuring expression changes of SOX2-activated (left) and SOX2-repressed (right) genes, identified by RNA-seq assays in KYSE140 cells with and without *SOX2* knockout, after e1 repression in KYSE140 cells ($n = 2$ biologically independent experiments). **g** Relationship between *SOX2* and e1-activated (left) and e1-repressed (right) genes across TCGA squamous cancer samples. *Z*-scores were generated as described in the "Methods" section. *P* values were derived from two-sided Spearman's correlation. Source data are provided as a Source Data file.

significant reductions in *LINC01206* expression (Supplementary Fig. 5b), which together with the observation that e6–e8 are next to *LINC01206* TSS suggests that they serve as promoter or promoter-proximal elements for this noncoding gene.

Given the predominant role of the e1 enhancer in *SOX2* regulation, we sought to examine structural variants targeting e1 in squamous cancers. We downloaded whole-genome sequencing (WGS) data for 113 squamous cancers from the Pan-Cancer Atlas of Whole-Genome (PCAWG) dataset[48,49]. GISTIC analysis of the segment data validated the focal amplification of the *SOX2*-e1 locus (Supplementary Fig. 6a). We identified 16 tumor samples with tandem duplications at the *SOX2*-e1 region (Supplementary Fig. 6b). Duplications in 12 of the cases contain both *SOX2* and e1. Interestingly, four tumor samples harbor duplications of only the enhancer region without the *SOX2* gene (Supplementary Fig. 6b), reminiscent of our previous findings regarding duplications of *MYC* and *KLF5* enhancers[5,28]. The presence of tandem duplications of just the enhancer region further highlights the importance of the e1 enhancer in squamous cancer.

**The e1 enhancer drives the activity of the e1–e5 enhancer cluster**. We next aimed to assess the functional link of e1 with the surrounding enhancers. Distal enhancers activate target gene expression by recruiting transcriptional coactivators such as the bromodomain protein BRD4 and the mediator complex that promote POL2 elongation[50]. ChIP-seq of the coactivator BRD4 showed that e1–e7 are enriched with BRD4 binding in ESSC KYSE140 cells. Repression of e1 decreased recruitment of BRD4 at not only e1 but also e2–e7 (Fig. 4a). H3K27ac HiChIP data showed that e1 physically interacts with the rest of the potential regulatory elements (Supplementary Fig. 7a), suggesting a structural basis for their interdependency. Globally, repression of e1 caused a clear reduction of BRD4 recruitment preferentially at high-confidence SOX2 binding sites (SOX2 ChIP-seq peaks containing SOX motifs) as compared to the other BRD4 sites in KYSE140 cells (Fig. 4b), which is likely due to the reduced abundance of the SOX2 transcription factor. We then performed BRD4 ChIP-seq in three additional squamous cancer cell lines LK2, NCI-H520, and HSC4 with and without e1 repression. The results consistently show that the activity of e1 is required for BRD4 recruitment at the surrounding enhancers (Fig. 4c).

In addition to e1, the e2–e7 elements are also enriched with SOX2 binding in KYSE140 cells (Supplementary Fig. 7b), which raised an important question of whether these enhancers are directly regulated by e1 or SOX2. To address this, we performed a rescue experiment by using the doxycycline-inducible *SOX2* expression system in KYSE140 cells with and without e1 repression. We performed BRD4 ChIP-qPCR by focusing on e1–e7 as they show significant BRD4 enrichment in KYSE140 cells. We found that ectopic expression of *SOX2* only rescued 27.0–32.4% of BRD4 binding at e2–e3 and 49.5–65.3% of the binding at e4–e5 (Supplementary Fig. 7c). In contrast, *SOX2* re-expression fully rescued the BRD4 binding at e6–e7 (Supplementary Fig. 7c). These results demonstrate that the enhancers e2–e5, but not e6–e7, are directly dependent on e1 to varying levels, defining an e1–e5 enhancer cluster.

**A combination of transcription factors including SOX2 itself contribute to the activity of the e1 enhancer**. We next sought to identify transcription factors that may contribute to e1 activity. Motif analysis of the e1 enhancer (chromatin-accessible region) identified DNA sequences recognized by multiple transcription factor families (Fig. 4d), most of which are distributed in regions that are highly conserved across species based on the PhastCons scores[51]. We then applied CRISPR/Cas9 to specifically disrupt the

DNA motifs within e1 and assessed their effects on *SOX2* expression (as illustrated in Fig. 4d and Supplementary Fig. 8a) in ESSC KYSE140 and LUSC LK2 cells. We observed >25% reductions of *SOX2* expression after disruptions of DNA motifs recognized by transcription factor families EHF, STAT, RUNX, SOX, and AP1 in KYSE140 cells and SNAIL, TCF, SOX, and AP1 in LK2 cells. The combinations of transcription factor motifs are different between these two tested cell lines, which is likely because that they represent two distinct types of squamous cancers. KYSE140 represents the classic *SOX2*-high and *TP63*-high squamous cancers, while LK2 was recently reported to represent a variant *SOX2*-high and *POU3F2*-high squamous cancer type that is enriched with neural signatures[38]. Despite the subtype difference, the transcriptional regulatory activity of e1 in both of the cell lines is dependent on the motifs recognized by SOX2 (SOX family motif) and the AP1 complex (Fig. 4e) that was previously indicated as a SOX2 cofactor[46], suggesting a positive feedback loop activating *SOX2* expression. For validation, we performed ChIP-qPCR in KYSE140 cells and showed that both SOX2 and FOSL1, a member of the AP1 complex, bind to the e1 enhancer, which was disrupted by CRISPR-mediated cutting of their respective motifs (Supplementary Fig. 8b). In addition, we also tested several additional SOX motifs in e2–e8 and the *SOX2* promoter, which showed that they have modest or minimal effect on *SOX2* expression (Supplementary Fig. 8c).

We then applied CRISPR/Cas9 to simultaneously disrupt SOX (2nd), AP1, RUNX, and STAT (2nd) motifs within the e1 enhancer in KYSE140 and SOX (2nd), AP1, SNAIL, and TCF motifs in LK2, which may either alter the nucleotides of the motif sequences or delete DNA fragments covering the motifs. We found that combinatorial cutting of the motifs resulted in more dramatic reductions in *SOX2* expression (76% for KYSE140 and 93% for LK2) as compared to individual motif disruptions, suggesting joint effects of the candidate transcription factors (Fig. 4e). Combinatorial disruptions of the motifs also caused reductions of BRD4 binding at not only e1 but also e2–e5 enhancers (Fig. 4f), which is consistent with the aforementioned finding that activity of the entire enhancer cluster is dependent on e1. Although some of the candidate functional motifs were also found in the e2–e5 enhancers, a full collection of the motifs was only observed at e1 (Fig. 4g), suggesting that combinatorial binding of the candidate transcription factors may determine the predominant role of e1 in the enhancer cluster.

**The coactivator BRD4 is required for *SOX2* activation, but is dispensable for the e1-*SOX2* loop**. Given the strong binding of the coactivator BRD4 at the e1 enhancer, we reasoned that the activity of e1 and the associated *SOX2* overexpression may be sensitive to BRD4 perturbation. To test this hypothesis, we applied the proteolysis targeting chimera (PROTAC) molecule ARV-771 to recruit the E3 ligase cereblon to degrade BRD4[52]. We found that 2 h of 0.5 μM ARV-771 treatment efficiently decreased the BRD4 protein level (Supplementary Fig. 9a) and removed the majority of BRD4 binding at e1 and its surrounding enhancers in KYSE140 cells (Fig. 5a). Indeed, the e1 enhancer is ranked as the top BRD4-bound regulatory element that is most sensitive to BRD4 degradation in KYSE140 cells (Fig. 5b). RNA-seq analysis showed that ~93% of *SOX2* expression was lost in KYSE140 cells after 6 h of 0.5 μM ARV-771 treatments (Fig. 5c). Comparable levels of reductions in *SOX2* expression were observed in five additional *SOX2*-high squamous cancer cell lines (Fig. 5d), suggesting common hypersensitivity of *SOX2* expression to BRD4 degradation. We also observed >50% reductions in the proliferation of KYSE140 and LK2 cells in response to 2 days of 0.5 μM ARV-771 treatments (Supplementary Fig. 9b).

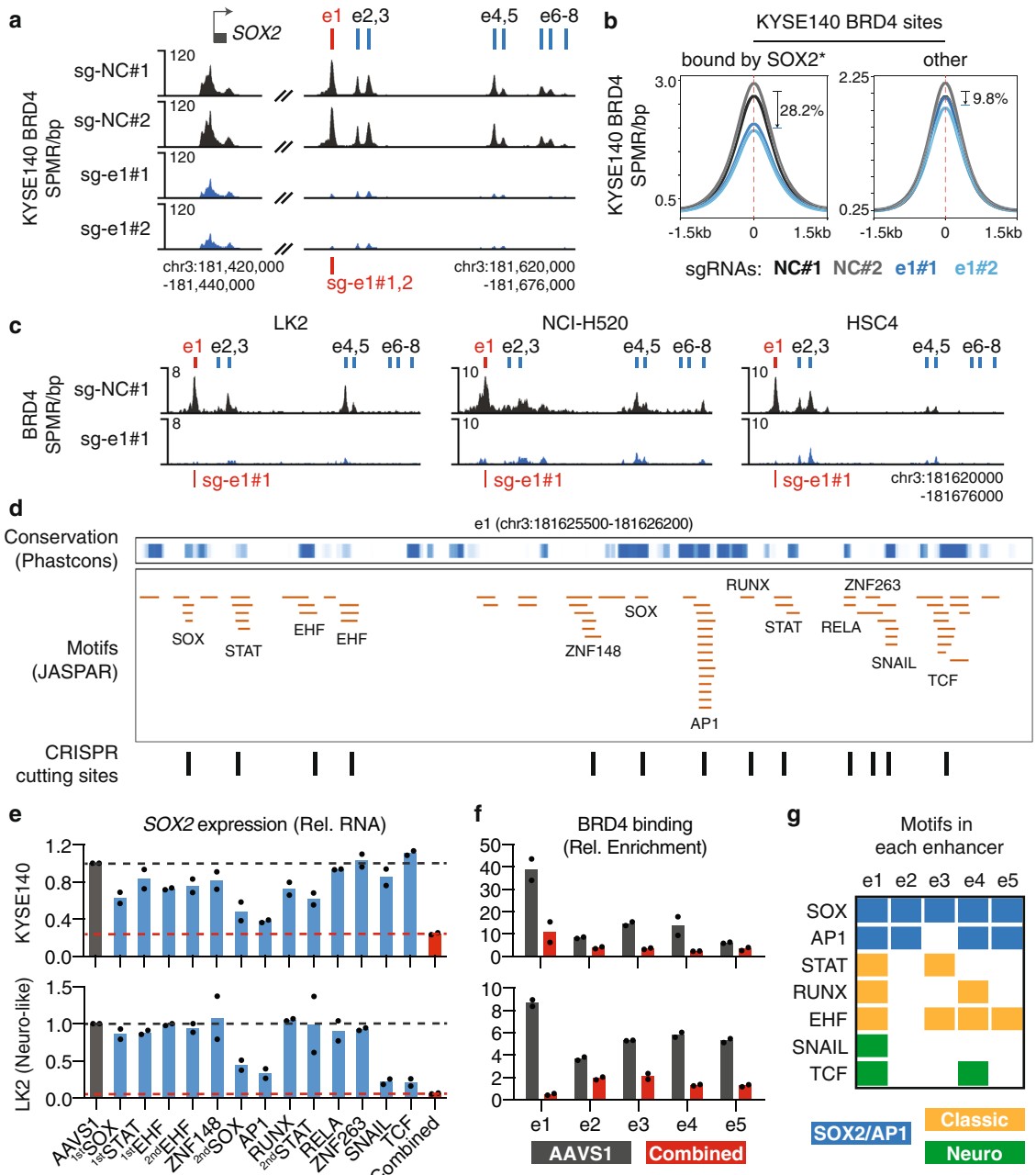

**Fig. 4 The e1 enhancer drives the activity of the entire enhancer cluster. a** BRD4 ChIP-seq profile at the *SOX2* locus in KYSE140 cells with and without e1 repression. **b** Averaged BRD4 ChIP-seq profile, across BRD4 sites that harbor high-confidence SOX2 binding (*SOX2 ChIP-seq peaks containing SOX motifs) or the other BRD4 sites, in KYSE140 cells with and without e1 repression. **c** BRD4 ChIP-seq profile at the e1–e8 locus in LK2, NCI-H520, and HSC4 cells with and without e1 repression. **d** Top: PhastCons scores (0:1 range) representing the conservation level of DNA sequences in the e1 enhancer. Middle: distribution of JASPAR DNA motifs identified in the e1 enhancer. Bottom: CRISPR cutting sites that overlap with the identified DNA motifs. **e** RT-qPCR measuring expression changes of *SOX2* in KYSE140 and LK2 cells after CRISPR-mediated disruption of each of the identified DNA motifs. The expression level was normalized to the sgAAVS1 control. *n* = 2 biologically independent experiments. *: combinatorial CRISPR cutting of SOX (2nd), AP1, RUNX, and STAT (2nd) motifs in KYSE140 cells, or SOX (2nd), AP1, SNAIL, and TCF motifs in LK2 cells. **f** ChIP-qPCR showing the relative enrichment of BRD4 at e1–e5 in KYSE140 and LK2 cells after combinatorial CRISPR cutting of the selected motifs. ChIP enrichment was normalized to DNA concentration of each sample (measured by Qubit) and then to sonicated genomic input. *n* = 2 biologically independent experiments. **g** Presence of the candidate functional motifs in the e1–e5 enhancers. Source data are provided as a Source Data file.

While we observed only ~35% reduction in *SOX2* expression after 2 h of 0.5 μM ARV-771 treatments in KYSE140 cells (Fig. 5e), we reasoned that most of the remaining signal may come from *SOX2* RNA that was already transcribed before the drug treatment. In order to assess the immediate effect of BRD4

degradation on *SOX2* transcription, we applied 4-thiouridine (4sU) to label the newly transcribed RNA, also known as nascent RNA, which was then captured by biotin pulldown and quantified by quantitative reverse transcription PCR (RT-qPCR). We showed that 2 h of ARV-771 treatments resulted in ~88% loss

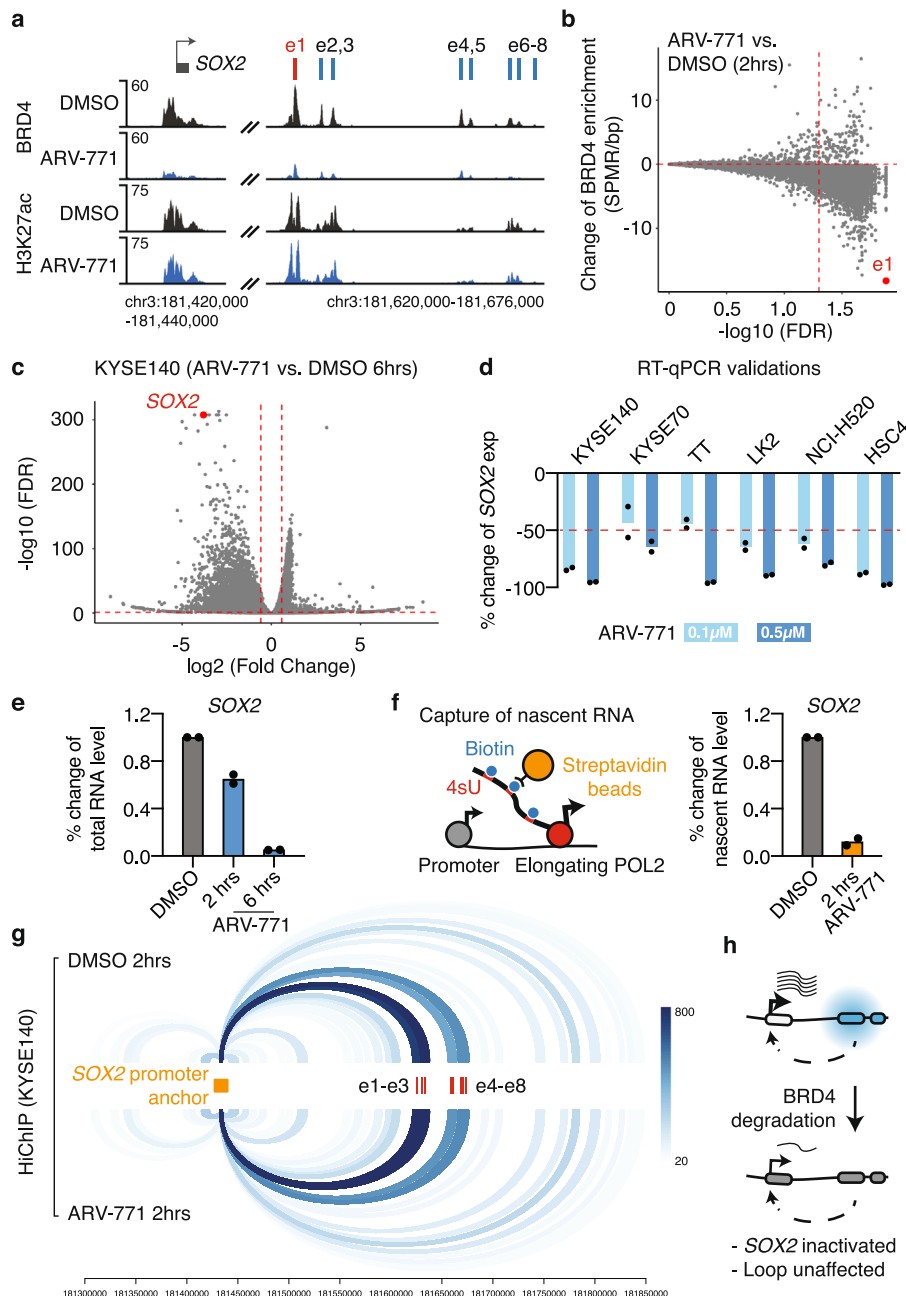

**Fig. 5 BRD4 is necessary for e1 activity, but dispensable for the e1-*SOX2* chromatin loop. a** ChIP-seq profile of BRD4 and H3K27ac at the *SOX2* locus in KYSE140 cells with 2 h of DMSO or 0.5 μM ARV-771 treatments. **b** The change of BRD4 ChIP-seq signal across all the BRD4 binding sites in KYSE140 cells after 2 h of 0.5 μM ARV-771 treatments, as compared to DMSO controls (two biological replicates). **c** RNA-seq results highlighting that *SOX2* is one of the top genes downregulated in KYSE140 cells after 6 h of 0.5 μM ARV-771 treatments. **d** RT-qPCR results showing relative expression changes of *SOX2* in six squamous cancer cell lines with 6 h of 0.1 or 0.5 μM ARV-771 treatments. The expression level was normalized to the DMSO controls. $n = 2$ biologically independent experiments. **e** RT-qPCR results showing relative expression changes of *SOX2* in KYSE140 cells after two and 6 h of 0.5 μM ARV-771 treatments. The expression level was normalized to the DMSO control. $n = 2$ biologically independent experiments. **f** Left: schematic illustrating the 4sU labeling assay that was applied to capture nascent RNA. Right: RT-qPCR of the captured nascent RNA transcribed from the *SOX2* gene in KYSE140 cells with 2 h treatments of DMSO or 0.5 μM ARV-771. RT-qPCR signal was normalized to the nascent RNA level of the *HPRT1* gene and then to the DMSO control. $n = 2$ biologically independent experiments. **g** H3K27ac HiChIP loops that are connected to the *SOX2* promoter in KYSE140 cells treated with 2 h of DMSO or 0.5 μM ARV-771. The color intensity corresponds to the number of PETs supporting each of the loops. **h** Schematic illustrating that ARV-771-mediated BRD4 degradation causes an acute and remarkable reduction in the e1 enhancer activity, but has little effect on the e1-*SOX2* loop. Source data are provided as a Source Data file.

of the nascent *SOX2* transcription, demonstrating an acute and remarkable response of *SOX2* transcription to BRD4 degradation (Fig. 5f).

We then sought to assess if BRD4 degradation affects the chromatin interaction between the e1 enhancer and the *SOX2* promoter. We performed HiChIP of H3K27ac in KYSE140 cells with 2 h of dimethyl sulfoxide (DMSO) or 0.5 μM ARV-771 treatment. H3K27ac serves as an ideal bait for the HiChIP capture in this experiment, as the enrichment of H3K27ac at the *SOX2*-e1 locus was barely affected by 2 h of ARV-771 treatment (Fig. 5a).

Surprisingly, despite the dramatic response of e1 activity and *SOX2* expression to BRD4 degradation, no appreciable change was observed for the chromatin interaction between e1 and the *SOX2* promoter (Fig. 5g), suggesting that the bromodomain protein BRD4 is dispensable for maintaining the chromatin loop (illustrated in Fig. 5h).

**Activation of e1 is sufficient to drive *SOX2* expression and rebuild the e1-*SOX2* chromatin loop**. We next aimed to investigate if activation of e1 is sufficient to drive *SOX2* expression. We selected two LUSC cell lines RERFLCAI and SKMES1 and one ESSC cell line TE1, all of which are associated with low *SOX2* expression (Supplementary Fig. 3b). We applied an improved CRISPR activation (CRISPRa) system, which utilized MS2 and PP7 RNA stem-loops to bring together multiple transcriptional activators such as VP64, p65, and HSF1 and the dCas9 protein[53]. We used two separate sgRNAs to recruit the CRISPRa complex to the e1 enhancer. Activation of e1 resulted in 8–146-fold increases of *SOX2* expression (averaged value of two separate sgRNAs) across the tested cell lines, demonstrating that e1 is sufficient for *SOX2* activation (Fig. 6a). Immunoblotting showed that e1 activation also increased SOX2 protein level in SKMES1 cells, which is comparable to that induced by promoter activation of the *SOX2* gene (Fig. 6b). In agreement with the aforementioned finding that *SOX2-OT* is a target gene of the SOX2 transcription factor, activation of e1 also caused upregulation of *SOX2-OT* expression in SKMES1 cells (Supplementary Fig. 10). As compared to e1, activation of e2–e8 elements have modest or minimal effects on *SOX2* expression, again highlighting the predominant role of e1. Activation of e6–e8 that are next to *LINC01206* TSS resulted in 10–45-fold increases of *LINC01206* expression, which agrees with their roles as a promoter or promoter-proximal elements for this noncoding gene.

RNA-seq analysis revealed that *SOX2* was the most significantly upregulated gene (fold change = 162; FDR = $8.8e^{-111}$) in SKMES1 cells after e1 activation (Fig. 6c). GSEA analysis showed that e1 activation in SKMES1 cells significantly induced SOX2-associated transcriptional programs that we identified from *SOX2* knockouts in the *SOX2*-high KYSE140 cell line. Indeed, e1 activation significantly upregulated SOX2-activated genes (normalized enrichment score = 1.43, P = 0.003) and downregulated SOX2-repressed genes (normalized enrichment score = −1.30, P = 0.009) (Fig. 6d). Finally, we performed H3K27ac HiChIP assays in SKMES1 cells with and without e1 activation, which revealed that e1 activation led to the formation of the e1-*SOX2* chromatin loop (Fig. 6e). We also observed increased chromatin interactions between *SOX2* and other enhancers surrounding e1, suggestive of a reconstitution of the chromatin architecture at the *SOX2* locus. Taken together, we show that activation of e1 is sufficient to bridge the e1 enhancer to the *SOX2* promoter, which results in transcriptional activation of the *SOX2* oncogene in squamous cancer cells (illustrated in Fig. 6f).

## Discussion

We and others have previously shown that overexpression of oncogenes can be driven by copy number amplifications of distal enhancers[5,20–28]. Here, we show that this phenomenon extends to lineage-specific enhancers in a cancer type-specific manner. Squamous cancers and gliomas selectively amplify enhancers located 3′ and 5′ to the *SOX2* gene, respectively, exhibiting a spatial switch of cancer type-specific copy number amplifications. The phenomenon is likely caused by the unique chromatin architecture surrounding the *SOX2* gene: (1) glioma- and squamous cancer-specific enhancers are distributed in two adjacent insulated neighborhoods demarcated by CTCF binding; (2) *SOX2*

resides right at the boundary of the two neighborhoods so that it has access to both. A recent study showed that copy number amplifications of oncogenes including *SOX2* may occur as different forms of structural events such as linear tandem duplications, chromosomal rearrangements, or extrachromosomal circular DNA[54]. We reveal that *SOX2* and distal enhancers that are looped to the *SOX2* promoter are often co-amplified in squamous cancers, suggesting a common transcriptional regulatory mechanism that may be shared by different structural forms of *SOX2* amplifications.

Lineage-specific oncogenes are known to be driven by condensed clusters of enhancer elements, namely super-enhancers or stretch enhancers[55,56,57], yet the hierarchical structures for most of these enhancer clusters remain largely unknown. We show that the enhancer cluster co-amplified with *SOX2* in squamous cancer is predominantly driven by a single enhancer e1, which aligns with previous findings of predominant enhancers in other enhancer clusters[58,59]. Within the *SOX2* enhancer cluster, all the remaining enhancers physically interact with e1 and are dependent on the activity of e1, but individually have a minimal or modest impact on *SOX2* expression. It is possible that some of the enhancers are redundant to each other in activating *SOX2*— repression of e1 collapses the entire enhancer cluster and thereby impairs the redundancy. Our work suggests that the predominant role of e1 may be driven by a series of squamous cancer-relevant transcription factors such as SOX2 itself, AP1, and potential family members of RUNX, STAT, SNAI, and TCF complexes. Identification of such predominant enhancers and their associated protein complexes will clarify mechanisms underlying the activation of lineage-specific oncogenes.

As transcription factors are difficult targets with small molecules, understanding the mechanisms underlying their transcriptional activation may imply alternative therapeutic strategies. This is particularly important for squamous cancers that are largely associated with copy number amplifications and the transcriptional activation of transcription factor genes such as *SOX2*, *TP63*, and *KLF5*[4,5,11–13,60]. While the encoded transcription factors are hard to be therapeutically targeted, enhancer activation may yield unique vulnerabilities for cancer cells driven by these oncogenes. We show that the activity of the *SOX2* enhancer is dependent on the SOX2 transcription factor itself, its potential cofactor AP1, and the transcriptional coactivator BRD4, representing a self-regulatory circuit that is normally hypersensitive to transcriptional inhibitors—a unique vulnerability that has been reported for other oncogenic transcription factors[61]. We show that PROTAC-mediated BRD4 degradation leads to an acute and dramatic reduction of *SOX2* transcription, suggesting an alternative strategy to target squamous cancers with *SOX2* activation, although the efficacy and specificity of such treatments require further preclinical investigations.

It remains elusive how enhancer–promoter loops are initiated and maintained. We show that CRISPR-mediated activation of the e1 enhancer is sufficient to rebuild the e1-*SOX2* loop, suggesting that enhancer activation is a prerequisite for initiating enhancer–promoter loops. On the other hand, despite the remarkable impact of BRD4 degradation on *SOX2* transcription, we find that it has little effect on maintaining the e1-*SOX2* chromatin loop. The observation agrees with recent findings of the *MYC* and *BCL2* loci in leukemia cells[62]. Similar findings have also been reported for the mediator complex, another important transcriptional coactivator[63,64]. These together suggest that enhancer activation may be dispensable for maintaining enhancer–promoter loops. Previous studies have shown that binding of several transcription factors such as CTCF, ZNF143, and YY1 to promoters or promoter-proximal regions is required for maintaining enhancer–promoter loops[36,37,65,66]. Indeed, we

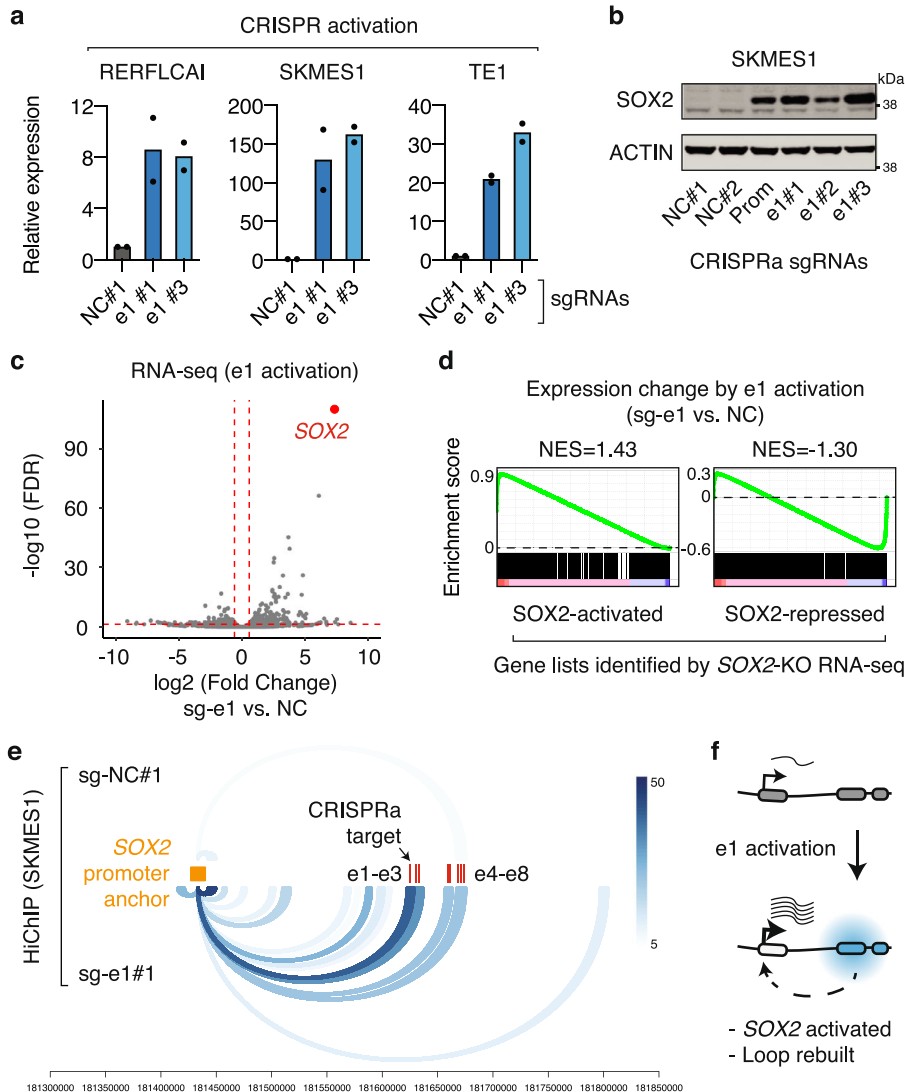

**Fig. 6 CRISPR-mediated e1 activation upregulates *SOX2* expression and rebuilds the e1-*SOX2* loop. a** RT-qPCR results showing the relative *SOX2* expression in three squamous cancer cell lines with and without e1 activation. The expression level was normalized to the negative control sg-NC#1. $n = 2$ biologically independent experiments for each sgRNA. **b** Immunoblots showing the protein level of SOX2 in SKMES1 cells with and without *SOX2* promoter activation or e1 enhancer activation. The immunoblotting experiment was repeated once independently with similar results. **c** RNA-seq results highlighting that *SOX2* is the top gene that is upregulated in SKMES1 cells after e1 activation. $n = 2$ biologically independent experiments. **d** GSEA analysis measuring expression changes of SOX2-activated or -repressed genes, identified by RNA-seq assays in KYSE140 cells with and without *SOX2* knockout, after e1 activation in SKMES1 cells. **e** H3K27ac HiChIP loops that are connected to the *SOX2* promoter in SKMES1 cells with and without CRISPR-mediated e1 activation (signal of two biological replicates were merged). The color intensity corresponds to the number of PETs supporting each of the loops. **f** Schematic illustrating that CRISPR-mediated e1 activation upregulates *SOX2* expression and rebuilds the e1-*SOX2* loop. Source data are provided as a Source Data file.

observed DNA recognition motifs of these transcription factors in the promoter-proximal region of *SOX2*. Future investigations are needed to determine whether and which enhancer-bound transcription factors play similar roles in promoter–enhancer interactions.

## Methods

**Cell lines**. Squamous cancer cell lines KYSE140, KYSE70, LK2, NCI-H520, HSC4, TE1, TE10, SKMES1, and RERFLCAI were obtained from the Broad Institute Cancer Cell Line Encyclopedia (CCLE) project[67,68]. The esophageal squamous cancer cell line TT was obtained from the Japanese Collection of Research Bioresources Cell Bank. Cells were grown in RPMI-1640 media supplemented with 10% fetal bovine serum and 1% of penicillin–streptomycin and tested negative for mycoplasma using the Lonza MycoAlert Detection kit. Cell lines were used for experiments after <3 months of passages. Cell line identities were verified by either SNP-array-based fingerprinting as previously described in the CCLE project[67,68] or short tandem repeat analysis.

**Analysis of TCGA and PCAWG datasets**. TCGA pan-cancer copy number segment dataset was downloaded from the National Cancer Institute Genomic Data Commons data portal (URL: https://gdc.cancer.gov/about-data/publications/pancanatlas). We performed GISTIC2[29] analysis using combined glioma data (LGG and GBM) and combined squamous cancer data (LUSC, HNSC, ESSC, and CESC) to call significantly focally amplified regions in the two cancer types. TCGA ATAC-seq data was published by Corces et al.[33] and downloaded from the NCI Genomic Data Commons data portal (URL: https://gdc.cancer.gov/about-data/publications/ATACseq-AWG). We used the published bigWig data for Integrative Genomics Viewer presentation and the normalized ATAC-seq insertion counts across the identified pan-cancer peak set for calculating Z-scores for each cancer type. TCGA-processed RNA-seq data was downloaded from the Firehose GDAC data portal of the Broad Institute (URL: http://gdac.broadinstitute.org/). For RNA expression correlation of *SOX2* and genes regulated by the e1 enhancer, we first calculated Z-scores for each tumor based on *SOX2* expression level (log 2 (RSEM + 1)) or the sum of the expression level of e1-regulated genes. One thousand six hundred and thirty-four e1-activated and 1391 e1-repressed genes (based on RNA-seq in KYSE140 cells with and without e1 repression: FDR < 0.05; fold

change > 1.5) were used for the analysis. We used the Z-scores for Spearman's correlation analysis to examine the relationship between the expression of SOX2 versus e1-regulated genes. Squamous cancer WGS-based copy number segment data and structural variants data were downloaded from the PCAWG data portal (URL: https://dcc.icgc.org/releases/PCAWG/).

**CRISPR-mediated enhancer repression and activation.** For enhancer repression, we first subcloned dCas9-KRAB-MeCP2 (Addgene, 110821) into BamHI–NheI sites of lenti-Cas9-Blast (Addgene, 52962) to generate a lentiviral dCas9-KRAB-MeCP2 vector. We then infected cells with the vector to stably express the dCas9-KRAB-MeCP2 fusion. The infected cells were selected with blasticidin (10 μg/ml) for at least 5 days. Enhancer-targeting sgRNAs were designed close to the summits of ATAC-seq peaks within the SOX2 enhancer cluster. We then infected the dCas9-KRAB-MeCP2 cells with LentiGuide-Puro (Addgene: 52963) carrying either previously published sgRNAs that have no recognition sites in the genome[4,28] or sgRNAs targeting the SOX2 enhancers. The infected cells were selected with puromycin (2 μg/ml) for at least 2 days before any molecular or cellular assays. For enhancer activation, cells were first infected with lenti-dCas9-VP64-Blast (Addgene: 61425) and selected with blasticidin (10 μg/ml) for at least 5 days. We then infected the dCas9-VP64 cells with pXPR502 (Addgene 96923) carrying either negative control or enhancer-targeting sgRNA, and selected the cells with puromycin (2 μg/ml) for at least 2 days. All sgRNA sequences are listed in Supplementary Table 2.

**Cell proliferation assays.** For cell proliferation, infected and selected cells were seeded at the same number (0.025 or 0.05 million) in a 6-well plate and counted after 6 days. For phenotype-rescue experiment, we first cloned SOX2 complementary DNA (cDNA) into the doxycycline-inducible expression vector pCW57.1-Puro (Addgene: 41393) and infected dCas9-KRAB-MeCP2 KYSE140 cells with pCW57.1-SOX2-Puro. We then used LentiGuide-Zeo (to avoid overlap of selection markers) to express sgRNAs. Cells were selected with zeocin (800 μg/ml) for 3 days and the same number of cells were then seeded with or without 1 μg/ml doxycycline. The cell culture media were changed with fresh doxycycline every other day before counting.

**CRISPR-mediated gene knockouts and DNA motif cutting.** We first generated Cas9-expressing cells by infecting cells with lenti-Cas9-Blast (Addgene: 52962). The infected cells were selected with blasticidin (10 μg/ml) for at least 5 days. We then infected the Cas9-expressing cells with LentiGuide-Puro carrying previously published sgRNAs targeting negative control regions sgAAVS1 and sg-Chr.2-2[4,69], the coding region of SOX2, or the transcription factor DNA binding motifs identified in the e1 enhancer. The infected cells were selected with puromycin (2 μg/ml) for at least 2 days before any experiments. All sgRNA sequences are listed in Supplementary Table 2.

**ChIP-seq and ChIP-qPCR.** ChIP-seq assays were performed as previously described[4]. Briefly, five million cells were crosslinked with 1% formaldehyde (diluted in 1× phosphate-buffered saline (PBS)) and lysed with Lysis Buffer I (5 mM PIPES pH 8.0, 85 mM KCl, 0.5% NP40) and then Lysis Buffer II (1× PBS, 1% NP40, 0.5% sodium deoxycholate, 0.1% sodium dodecyl sulfate) supplemented with protease inhibitors. Chromatin extract was sonicated with QSonica Q800R (pulse: 30 s on/30 s off; sonication time: 20 min; amplitude: 70%) and immuno-precipitated with antibodies premixed with Dynabeads A and G. Antibodies: H3K27ac (Abcam, ab4729, rabbit polyclonal, 4 μg/ChIP), BRD4 (Bethyl, A301-985A100, rabbit polyclonal, 4 μg/ChIP), SOX2 (R&D Systems, AF2018, goat polyclonal, 4 μg/ChIP), CTCF (Cell Signaling, 2899, rabbit polyclonal, 10 μl/ChIP), FOSL1 (Cell Signaling, 5281, rabbit monoclonal, clone D80B4, 10 μl/ChIP), Cas9 (Diagenode, C15310258, rabbit polyclonal, 4 μg/ChIP). ChIP-seq libraries were prepared using NEBNext DNA Ultra II library prep kit and NEBNext Multiplex Oligos for Illumina (96 Unique Dual Index Primer Pairs), and sequenced by Illumina NovaSeq. For ChIP-qPCR assays, we designed primers targeting individual SOX2 enhancers and used sonicated genomic DNA to normalize primer efficiency variance. All the qPCR primers are listed in Supplementary Table 2.

We used Bowtie2[70] to align sequencing reads to the hg19 human genome, Samtools[71] to sort and index the aligned reads, and MACS2[72] to calculate signal per million reads (SPMR) and to call significant ChIP-seq peaks (q value < 0.05). For super-enhancer analysis, we used the Homer pipeline[73] to identify super-enhancers based on H3K27ac ChIP-seq signal, and then used bedtools[74] to intersect super-enhancers called from SOX2-high squamous cancer cell lines, which identified the shared super-enhancer region in the SOX2 locus. For enhancer comparison between human and mouse cells, we applied the UCSC LiftOver tool[43] to identify mouse genomic regions that are conserved to human squamous cancer SOX2 enhancers and then compared these regions to mouse ESC Sox2 enhancers. For measuring the effect of e1 repression on global BRD4 ChIP-seq signal, we first categorized BRD4 binding sites (BRD4 ChIP-seq peaks identified in KYSE140 cells with sg-NC#1) into two groups based on whether or not they overlap with high-confidence SOX2 binding sites (SOX2 ChIP-seq peaks containing SOX motifs). We used deepTools[75] to present averaged BRD4 ChIP-seq profile across these two groups of BRD4 sites in KYSE140 cells with and without e1 repression. To calculate

the change of BRD4 ChIP-seq signal at individual enhancers in KYSE140 cells post ARV-771 treatment, we used the UCSC tool "bigWigAverageOverBed"[76] to calculate the SPMR value for each BRD4 binding site (BRD4 ChIP-seq peaks identified in KYSE140 cells treated with DMSO) and then performed t tests to compare the values from cells with DMSO or ARV-771 treatments.

**RNA-seq and RT-qPCR.** Total RNA was extracted using the Zymo Quick-RNA miniprep kit with on-column DNase treatments. mRNA was then purified using the NEBNext Poly-A mRNA Magnetic Isolation Module. RNA-seq libraries were prepared using NEBNext Ultra Directional RNA library prep kit and NEBNext Multiplex Oligos for Illumina (96 Unique Dual Index Primers), and then sequenced by Illumina NovaSeq. Sequencing reads were aligned to the hg19 human genome using Bowtie2[70]. Expression level (read counts) for each GENCODE gene was quantified with RSEM[77]. We applied the edgeR package[78] to normalize the read counts and to perform differential expression analysis. We applied the cut-off of FDR < 0.05 and fold change > 1.5 to identify genes that are significantly down- or upregulated after SOX2 knockout, e1 repression, or e1 activation. To compare gene expression changes caused by SOX2 knockouts versus e1 repression or activation, we ranked SOX2-regulated genes based on their edgeR FDR values and selected the top 1000 SOX2-activated or -repressed genes as "gene sets," which we used for GSEA analysis to assess if they are down- or upregulated after e1 repression or activation. For RT-qPCR, the extracted RNA was first converted into cDNA with NEB LunaScript SuperMix kit and then processed with real-time PCR using the NEB Luna Universal qPCR Master Mix on a Bio-Rad CFX96 qPCR instrument. The signal of qPCR was normalized to the internal reference genes ACTB or HPRT1. Primers used for RT-qPCR were listed in Supplementary Table 2.

**HiChIP and loop calling.** HiChIP was performed based on the previously published protocol[44] with several modifications as previously described[4]. Briefly, crosslinked chromatin was first digested with the MboI restriction enzyme, filled with dCTP, dGTP, dTTP, and biotin-labeled dATP, ligated with T4 DNA ligase, and sonicated to achieve an average fragment length of ~1 kb. Antibodies of H3K27ac (Abcam, ab4729, rabbit polyclonal, 7.5 μg/HiChIP; Active Motif, 39133, rabbit polyclonal, 7.5 μg/HiChIP) were used to capture DNA fragments with potential regulatory activity. The streptavidin C1 magnetic beads were used to enrich DNA fragments that were successfully ligated. HiChIP libraries were generated with the Illumina Tagment DNA Enzyme and Buffer kit and sequenced with Illumina NextSeq or NovaSeq.

The sequencing reads were aligned to the hg19 human genome using the HiC-Pro pipeline[79]. For calling chromatin loops, we first generated a union of H3K27ac sites by merging H3K27ac ChIP-seq binding sites (broad peaks) identified in SOX2-high squamous cancer cell lines KYSE140, TT, TE10, LK2, and NCI-H520. We then used these sites as "anchors" and counted the number of PET spanning these anchors using the Hichipper pipeline[80]. For presentation, we selected the significant loops (PETs ≥ 5 and FDR < 0.05) that are connected to the anchor overlapping with the SOX2 transcription start site. For presenting HiChIP data from SKMES1 cells with and without CRISPR-mediated e1 activation, two biological replicates were merged.

**Motif analysis.** We used ATAC-seq signal from TCGA squamous cancer samples and BRD4 ChIP-seq signal from KYSE140 cells to narrow down DNA coordinates of candidate SOX2 enhancer regions. We then used the FIMO software[81] with a threshold of P value <10^{-4} to identify transcription factor binding motifs from the JASPAR motif database[82] that are present in the SOX2 enhancers.

**4sU labeling of nascent RNA.** 4sU labeling of nascent RNA was performed as previously described[83] with minor modifications. Two million cells per 10 cm dish were seeded one day prior to labeling. Cells were first treated with 0.5 μM ARV-771 or the same volume of DMSO control for 2 h at 37 °C and then treated with 200 μM 4sU (Sigma Aldrich, T4509) or the same volume of DMSO for 20 min at 37 °C. Per condition, cells were harvested and processed with TRIzol and Zymo Quick-RNA miniprep purification. Twenty micrograms of purified RNA was mixed with 5 μg of MTSEA biotin-XX (Biotium, Cat#90066) in 400 μl of Biotinylation buffer (10 mM HEPES pH 7.5, 1 mM EDTA, and 20% dimethylformamide) and incubated for 2 h with rotation in the dark. Free biotin was then removed with Zymo RNA Clean & Concentrator kit. One hundred microliters of Dynabeads MyOne Streptavidin C1 (Thermo Fisher) was added to RNA and incubated in 200 μl Biotin binding buffer (10 mM Tris-HCl pH 7.5, 2 mM EDTA, 200 mM NaCl, 0.02%Tween-20) for 1 h with rotation. Beads were washed three times with the Biotin binding buffer. RNA was eluted sequentially with 5% β-mercaptoethanol for 15 min at room temperature and 100% β-mercaptoethanol for 5 min at 50 °C. The combined RNA elute was purified with Zymo RNA Clean & Concentrator kit. The abundance of SOX2 nascent transcription was quantified using RT-qPCR and normalized to HPRT1 nascent transcription. All the qPCR primers are listed in Supplementary Table 2.

**Immunoblotting.** Cells were lysed with NP40 lysis buffer (1% NP40, 150 mM NaCl, and 50 mM Tris-HCl, pH 8.0) supplemented with protease inhibitors and sonicated with QSonica Q800R (pulse: 30 s on/30 s off; sonication time: 2 min;

amplitude: 50%). Protein extract was denatured with LDS sample buffer (Thermo Fisher) supplemented with 20 mM dithiothreitol, separated in 4–12% NuPage Bis-Tris gel (Thermo Fisher), and transferred to nitrocellulose membranes. For immunoblotting, we used primary antibodies of SOX2 (Cell Signaling, 3728, rabbit monoclonal, clone C70B1, 1:1000 dilution), ACTIN (Santa Cruz, sc-47778, mouse monoclonal, clone C4, 1:2500 dilution), and BRD4 (Bethyl, A301-985A100, rabbit polyclonal, 1:1000 dilution), and secondary antibodies of goat anti-rabbit IRDye 800CW (LI-COR, 925-32211, 1:10,000 dilution), and goat anti-mouse IRDye 680CW (LI-COR, 925-68070, 1:10,000 dilution). In this study, we independently validated the specificity of the SOX2 antibody by using cells with and without ectopic expression of *SOX2* cDNA (Supplementary Fig. 4b) or cells with and without CRISPR-mediated *SOX2* knockout (Supplementary Fig. 4c). Immunoblotting images were taken on an LI-COR instrument following the manufacturer's instructions. For Figs. 3c and 6b and Supplementary Fig. 4b–c, we blotted the same membranes with primary antibodies of different species (SOX2: rabbit; ACTIN: mouse) and then LI-COR fluorescent secondary antibodies. For Supplementary Fig. 9a, we cut the same membrane in half and blotted BRD4 and ACTIN separately. The uncropped and unprocessed scans of the immunoblots were included in the Source data.

**In vivo xenograft assays**. All animal experiments were conducted in accordance with procedures approved by the institutional Animal Care and Use Committee at the Dana-Farber Cancer Institute, in compliance with NIH guidelines. For subcutaneous implantation, 2.5 million LK2 cells with and without CRISPR-mediated e1 repression were resuspended in 200 μl mixture (1:1 Matrigel:media) and injected into the flanks of female nude mice (6–8 weeks old, Nu/Nu; Jackson Laboratory). All the mice were housed in pathogen-free environment, with 12 h of light and 12 h of dark cycles, 18–21 °C, and 40–60% humidity. Mice were examined every 3–4 days, and tumor length and width were measured using calipers. The maximal tumor size permitted by our ethics committee is 2 cm (in any dimension). None of the tumors in our study exceeded the maximal limit. Tumor volume was calculated using the following formula: $(length \times width^2) \times 0.5$.

**Reporting summary**. Further information on research design is available in the Nature Research Reporting Summary linked to this article.

## Data availability

TCGA publicly available copy number and ATAC-seq data were downloaded from NCI Genomic Data Commons data portal (copy number URL: https://gdc.cancer.gov/about-data/publications/pancanatlas; ATAC-seq URL: https://gdc.cancer.gov/about-data/publications/ATACseq-AWG). TCGA publicly available RNA-seq data were downloaded from Broad Institute GDAC data portal (URL: http://gdac.broadinstitute.org/). PCAWG publicly available whole-genome sequencing data was downloaded from the PCAWG data portal (URL: http://gdac.broadinstitute.org/). The H3K27ac ChIP-seq publicly available data used in this study were downloaded from the Gene Expression Omnibus (GEO) series GSE137461 (for LK2, NCI-H520, and CALU1 cells), GSE88976 (for TT and TE10 cells), GSE16256 (for hESC), and GSE31039 (for mESC). The ChIP-seq, HiChIP, and RNA-seq data generated in this study have been deposited to GEO under the series GSE166234. The remaining data are available within the Article, Supplementary information, or Source Data file. Source data are provided with this paper.

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

## Acknowledgements

We thank Trudy Oliver (HCI) for useful discussions. S.D.B. is supported by the Canadian Institutes of Health Research (CIHR) and the Cancer Research Society (CRS). S.D.B. is the recipient of a Chercheur-boursier Junior 1 award from the Fonds de Recherche du Québec—Santé (FRQS), the Thomlinson award from McGill University, and the Dr. Ray Chiu distinguish scientist in surgical research award from the Montreal General Hospital Foundation. X.C. is supported by the National Natural Science Foundation of China (grant numbers: 82073637 and 82122060).

## Author contributions

D.K.A.R. and K.L.M. contributed equally. Y.L., S.D.B., and X.Z. designed the research and wrote the manuscript. Y.L., Z.W., J.Z., D.K.A.R., K.L.M., E.A.-J., and X.Z. performed the biological experiments. Y.L., Z.W, J.Z., Y.Y., A.M.T., A.D.C., S.D.B., and X.Z. performed the computational and statistical analyses. X.Y., K.E.V., J.G., P.S.C., X.C., A.J.B., S.D.B., and X.Z. supervised the research and reviewed and revised the manuscript. K.E.V., J.G., P.S.C., and X.C. provided technical and material support.

## Competing interests

E.A.-J. is employed at Recursion Pharmaceuticals. K.E.V. is a cofounder and consultant for Kailos Genetics. A.M.T. receives research funding from Ono Pharmaceutical. A.D.C. receives research funding from Bayer. A.J.B. receives research funding from Bayer, Merck, and Novartis, serves as a consultant to Earli and HelixNano, and is a cofounder of Signet Therapeutics. The other authors declare no competing interests.
