## [Peer Review File · Nature Communications]

A predominant enhancer co-amplified with the SOX2 oncogene is necessary and sufficient for its expression in squamous cancerReviewers' Comments:

Reviewer #1:

Remarks to the Author:

Liu et al. provide detailed analysis of SOX2 enhancers in squamous tumors, demonstrate that several enhancers are often amplified in squamous tumors and strongly regulate the expression of SOX2. In particular they focus on a key enhancer (e1) that is both required and sufficient for high expression of SOX2. This is a very elegant study that both shed light on the transcriptional regulation of SOX2 as a model for lineage-specific oncogenes, as well as provide insights into the regulatory networks that drive squamous tumors and how to potentially target them. However, there are two points that needs to be revised (see below). I recommend that the paper should be accepted for publication after these issues will be addressed.

1. The authors claim that e1 regulates the activity of the enhancer cluster and suggest this is mediated by interactions between e1 and the other enhancers. To support this claim, they show that silencing e1 disrupts BRD4 recruitment to the other enhancers as well. However, since silencing e1 downregulates SOX2, and since SOX2 is likely required for the activity of enhancers e2-e8, it seems likely that loss of e2-e8 is due to loss of SOX2. The authors should clarify whether this is a direct effect or not, for example by re-expressing SOX2 in the e1 repressed cells to test whether this restores BRD4 binding.

2. It is not clear what is the function of enhancers e2-e8 and whether they really regulate SOX2. Silencing these enhancers seem to have no effect on SOX2 expression. Does it affect the expression of other genes? Are there interactions between these enhancers and other promoters? Does activating these enhancers in SOX2-low cells effect SOX2 expression? Or the expression of other genes? The authors should clarify what is the function of these enhancers and why do they refer to them as enhancers.

Thank you,
Yotam Drier.

Reviewer #2:

Remarks to the Author:

The article reported an interesting epigenetic features of the lineage-specific SOX2 oncogene amplifications in squamous cell carcinoma. Through various sequencing analyses and CRISPR-mediated functional studies, the authors proven the critical role of a single enhancer e1 in driving SOX2 expression and the BRD4 inhibition as potential therapeutic strategy for SOX2 amplified squamous cell carcinoma.

The study is well-designed and comprehensive. I would like to suggest the authors to clarify some minor points about the SOX2 amplicon:

-In addition to the SNP-array-based copy number data, are there any whole genome sequencing or whole-exome sequencing data to support the focal SOX2 amplifications defined in this study? If the WGS dataset available, the SV in the e1-e8 regions should be examined.

-Since Sox2 is overlapped with the intron region of the SOX2-OT gene, the location of SOX-OT should be indicated in the Figures, e.g. Figure1 and 2. Supplementary Figure 2.

-The correlation of SOX2-OT expression and amplification in the squamous cancers and glioma should also been examine.

-The location of all potential TF binding sites of SOX2 and that included in the CRISPR cutting on the promoter and enhancer (e1 to e8) should be indicated in the figures.

REVIEWER COMMENTS

Reviewer #1, expert in 3D epigenomics and enhancers (Remarks to the Author):

Liu et al. provide detailed analysis of SOX2 enhancers in squamous tumors, demonstrate that several enhancers are often amplified in squamous tumors and strongly regulate the expression of SOX2. In particular they focus on a key enhancer (e1) that is both required and sufficient for high expression of SOX2. This is a very elegant study that both shed light on the transcriptional regulation of SOX2 as a model for lineage-specific oncogenes, as well as provide insights into the regulatory networks that drive squamous tumors and how to potentially target them. However, there are two points that needs to be revised (see below). I recommend that the paper should be accepted for publication after these issues will be addressed.

We appreciate the reviewer’s positive comments on our work. We have now performed additional experiments and analyses to address the two important points that the reviewer raised.

1. The authors claim that e1 regulates the activity of the enhancer cluster and suggest this is mediated by interactions between e1 and the other enhancers. To support this claim, they show that silencing e1 disrupts BRD4 recruitment to the other enhancers as well. However, since silencing e1 downregulates SOX2, and since SOX2 is likely required for the activity of enhancers e2-e8, it seems likely that loss of e2-e8 is due to loss of SOX2. The authors should clarify whether this is a direct effect or not, for example by re-expressing SOX2 in the e1 repressed cells to test whether this restores BRD4 binding.

The reviewer raised an important point regarding whether or not the e1 enhancer directly regulates the activity of e2-e8. Firstly, we would like to emphasize that among the 8 elements, only e1-e5 exhibit consistent BRD4 enrichment across the four tested SOX2-high squamous cancer cell lines. The e6-e7 elements show BRD4 enrichment only in KYSE140 cells, while e8 shows BRD4 enrichment only in NCI-H520 cells. We selected the KYSE140 cell line as our model, as it has BRD4 binding at seven out of the eight elements (Figure S7B). SOX2 ChIP-seq data shows that SOX2 also binds to e1-e7 in KYSE140 cells (Figure S7B). The e8 element, which has no significant enrichment of BRD4 (based on peak calling), is also not bound by SOX2.

As suggested by the reviewer, we then performed a rescue experiment to clarify the relationship of e1 and the other elements. We used the Doxycycline-inducible SOX2 expression system – the same system that we used for the cell proliferation phenotype-rescue assay in the study (Figure S4B). We performed BRD4 ChIP-qPCR and measured the relative change of BRD4 binding at e1-e7 after e1 repression in KYSE140 cells with and without SOX2 re-expression. The new results (Figure S7C) show that SOX2 re-expression:

1. failed to restore BRD4 binding at e1, which is most likely caused by the constant repression of e1 by the CRISPRi system in this experiment.
2. only restored of 27.0%-32.4% of BRD4 binding at the enhancers e2-e3, suggesting that they are, for the most part, directly regulated by e1.
3. restored 49.5%-65.3% of BRD4 binding at e4-e5, suggesting they are partially regulated by e1.
4. fully restored BRD4 binding at e6-e7, suggesting that their activity is regulated by SOX2 rather than the e1 enhancer.

Thanks to the reviewer’s suggestion, these new data have clarified the relationship between e1 and the other regulatory elements, which defines an e1-e5 enhancer cluster that is directly driven by e1. We have now included these data in **Figure S7** and described the results in **page 8, paragraph 3**: “In addition to e1, the e2-e7 elements are also enriched with SOX2 binding in KYSE140 cells (Figure S7B), which raised an important question of

Figure S7

Figure S7 B. ChIP-seq of BRD4 and SOX2 signal at the e1-e8 locus. *: based on peak calling, e8 has no significant enrichment of BRD4 in KYSE140 cells. **C.** BRD4 ChIP-qPCR results showing the % change of BRD4 binding at e1-e7 after e1 repression in KYSE140 cells with and without Doxycycline-induced ectopic expression of SOX2. P values are derived from t-tests: *<0.05; **<0.01; ***<0.001.

whether these enhancers are directly regulated by e1 or SOX2. To address this, we performed a rescue experiment by using the Doxycycline-inducible *SOX2* expression system in KYSE140 cells with and without e1 repression. We performed BRD4 ChIP-qPCR by focusing on e1-e7 as they show significant BRD4 enrichment in KYSE140 cells. We found that ectopic expression of *SOX2* only rescued 27.0%-32.4% of BRD4 binding at e2-e3 and 49.5%-65.3% of the binding at e4-e5 (Figure S7C). In contrast, *SOX2* re-expression fully rescued the BRD4 binding at e6-e7 (Figure S7C). These results demonstrate that the enhancers e2-e5, but not e6-e7, are directly dependent on e1 to varying levels, defining an e1-e5 enhancer cluster."

2. It is not clear what is the function of enhancers e2-e8 and whether they really regulate SOX2. Silencing these enhancers seem to have no effect on SOX2 expression. Does it affect the expression of other genes? Are there interactions between these enhancers and other promoters? Does activating these enhancers in SOX2-low cells effect SOX2 expression? Or the expression of other genes? The authors should clarify what is the function of these enhancers and why do they refer to them as enhancers.

Based on the reviewer's suggestion, we have now performed a series of analyses and experiments to define the function of e1-e8.

1. We first analyzed our H3K27ac HiChIP results in the five squamous cancer cell lines KYSE140, KYSE70, TT, LK2 and NCI-H520 to assess chromatin interactions between e1-e8 and gene promoter regions. The e1-e8 elements reside in four separate HiChIP anchors (e1, e2-e3, e4-e5, and e6-e8) that we previously identified in the study. We used the cutoff of paired-end tags (PETs) ≥ 5 and $FDR < 0.05$ to select significant chromatin interactions in each of the cell lines. We found that, in addition to the *SOX2* promoter, the promoters of protein-coding genes *FXR1*, *ATP11B* and the noncoding genes *SOX2-OT* (the P3 short isoform) and *LINC01206* have significant chromatin interactions with at least one of the four HiChIP anchors in two or more of the tested cell lines (Figure S5A). In addition, the e6-e8 elements are within 5kb +/- of TSS of the *LINC01206*, suggesting that they serve as the promoter or promoter proximal elements of the noncoding gene.

2. We then compared the number of PETs (representing chromatin interaction strength) between promoter regions of the candidate genes and the four HiChIP anchors. The analyses show that among these gene promoters, the *SOX2* promoter has the strongest interactions with these anchors in each of the tested cell lines (Figure S5A).

Figure S5

Figure S5 A. Number of PETs that connecting promoter regions of the candidate genes and the four HiChIP enhancer anchors. Note: The *SOX2-OT* gene has three alternative promoters - only the P3 promoter is connected to the enhancer anchors in two of the cell lines. N.S.: no significant loops. **B.** Expression change (%) of each of the candidate genes in KYSE140 cells after individual repression of e1-e8. Expression level is normalized to the negative control (sg-NC#1). **C.** Expression change (%) of the endogenous *SOX2* and *SOX2-OT* after e1 repression in KYSE140 cells with and without ectopic expression of *SOX2*. Primers targeting the *SOX2* 3'UTR region were used to distinguish the endogenous and ectopic *SOX2*. P value is derived from t-test: *** < 0.001 . **D.** *SOX2* ChIP-seq profile at the *SOX2-OT* (P3) promoter isoform locus.

- For functional assessment, we first performed CRISPRi assays in KYSE140 cells to repress each of the e1-e8 elements and tested their effects on expression of the five candidate genes (Figure S5B). In addition to *SOX2*, we found that e1 repression also caused a remarkable reduction in expression of the noncoding gene *SOX2-OT* (Figure S5B), which we have later found to be an indirect effect (see point #4 below). The e2-e5 elements individually have modest or minimal effects on expression of the candidate genes. As expected, repression of e6-e7, which are at the promoter region of *LINC01206*, caused significant reductions in expression of the noncoding gene (Figure S5B). Although e8 has little regulatory potential in KYSE140 cells (based on BRD4 and H3K27ac ChIP-seq signal), recruiting the dCas9-KRAB-MeCP2 repressor complex to e8 still resulted in a marked reduction of *LINC01206* expression. This is likely because that e8 is right downstream of the *LINC01206* TSS – the repressor complex at e8 may block POL2 elongation, a phenomenon that has been previously reported (Cho et al., 2018).
- One additional question raised from the CRISPRi experiment is whether or not the noncoding gene *SOX2-OT* is another direct gene target of e1. To address this, we made use of the Doxycycline-inducible *SOX2* expression system and found that re-expression of *SOX2* rescued expression of *SOX2-OT* that was decreased by e1 repression. In contrast, *SOX2* re-expression failed to rescue expression of the endogenous *SOX2* (RT-qPCR primers targeting the *SOX2* 3'UTR region were used to distinguish endogenous and ectopic *SOX2* cDNAs) (Figure S5C). In addition, we observed *SOX2* binding at and near the promoter region of the *SOX2-OT* gene (Figure S5D). These results demonstrate that *SOX2-OT* is a target of the *SOX2* transcription factor, rather than a direct target of the e1 enhancer.

- As suggested by the reviewer, in addition to CRISPRi, we also performed CRISPRa experiments to activate each of the e1-e8 elements in the *SOX2*-low squamous cancer cell line SKMES1 (Figure S10). The results largely agree with our findings from the CRISPRi experiments. Among the eight elements, activation of e1 resulted in the strongest upregulation of the *SOX2* gene. We showed that e1 activation also upregulated expression of *SOX2-OT*, which we have shown to be a target gene of the *SOX2* transcription factor (see point #4). In addition, activation of e6-e8, all within 5kb +/- of TSS of *LINC01206*, up-regulated expression of this noncoding gene to varying levels.

Figure S10

Figure S10: Expression fold change of *FXR1*, *SOX2-OT* (P3 promoter isoform), *SOX2*, *LINC01206*, and *ATP11B* in SKMES1 cells after CRISPR-mediated activation of e1-e8. The expression levels are normalized to the sg-NC#1 negative control.

In summary, thanks to the reviewer's question, we have now determined the function of e1-e8 elements. The e1 enhancer directly regulates *SOX2* and indirectly regulates *SOX2-OT*. The e2-e5 enhancers, which are partially dependent on e1, individually have modest or minimal effects on expression of *SOX2* or the other surrounding genes. The e6-e8 elements serve as the promoter or promoter-proximal elements for the *LINC01206* long-noncoding RNA gene. These new results are now included in **Figure S5** and **S10** and described in the following paragraphs:

Page 7, paragraph 2: "We then went on to test if e1 and the surrounding enhancers directly regulate any other genes in addition to *SOX2*. We analyzed the HiChIP data in *SOX2*-high squamous cancer cell lines by focusing on HiChIP anchors that harbor the e1-e8 elements (four anchors in total). We identified four additional candidate coding and noncoding genes *FXR1*, *ATP11B*, *SOX2-OT* and *LINC01206* – the promoter region of each gene interacts with at least one of the enhancer anchors in two or more of the five tested cell lines (Figure S5A). Among them, the *SOX2* promoter has the strongest interactions with these enhancer anchors. We then performed CRISPRi assays in KYSE140 to assess the effects of e1-e8 on these candidate genes. In addition to *SOX2*, e1 repression also decreased *SOX2-OT* expression (Figure S5B). However, ectopic expression of *SOX2*, which had no effect on the decreased endogenous *SOX2* expression, rescued the decreased *SOX2-OT* expression (Figure S5C). This result, together with our observation of several *SOX2* binding sites at or next to *SOX2-OT* promoter region (Figure S5D), suggests that *SOX2-OT* is directly regulated by *SOX2* but not e1. Repression of e6-e8 caused significant reductions in *LINC01206* expression (Figure S5B), which together with

the observation that e6-e8 are next to *LINC01206* TSS suggests that they serve as promoter or promoter-proximal elements for this noncoding gene.”

Page 10, paragraph 4: “In agreement with the aforementioned finding that *SOX2-OT* is a target gene of the SOX2 transcription factor, activation of e1 also caused upregulation of *SOX2-OT* expression in SKMES1 cells (Figure S10). As compared to e1, activation of e2-e8 elements have modest or minimal effects on *SOX2* expression, again highlighting the predominant role of e1. Activation of e6-e8 that are next to *LINC01206* TSS resulted in 10 to 45-fold increases of *LINC01206* expression, which agrees with their roles as promoter or promoter-proximal elements for this noncoding gene.”

Based on these observations, we have also now referred to e6-e8 as “promoter or promoter-proximal elements of *LINC01206*”, instead of “enhancers”, in the manuscript.

Reviewer #2, expert in squamous cancer genomics and models (Remarks to the Author):

The article reported an interesting epigenetic features of the lineage-specific *SOX2* oncogene amplifications in squamous cell carcinoma. Through various sequencing analyses and CRISPR-mediated functional studies, the authors proven the critical role of a single enhancer e1 in driving *SOX2* expression and the BRD4 inhibition as potential therapeutic strategy for *SOX2* amplified squamous cell carcinoma.

The study is well-designed and comprehensive. I would like to suggest the authors to clarify some minor points about the *SOX2* amplicon:

We appreciate the reviewer's positive comments on our manuscript. We have now performed additional analyses and experiments to address the reviewer's questions and also improved our figure annotations based on the reviewer's suggestions.

-In addition to the SNP-array-based copy number data, are there any whole genome sequencing or whole-exome sequencing data to support the focal *SOX2* amplifications defined in this study? If the WGS dataset available, the SV in the e1-e8 regions should be examined.

We would like to thank the reviewer for the suggestion, since it provided additional support for our analysis and findings. We were able to access whole genome sequencing (WGS) data for 113 of squamous cancers that were part of the Pan-Cancer Atlas of Whole Genome (PCAWG) project. The released dataset contains two types of data: 1) copy number segments called based on WGS read coverage, 2) structural variants called based on chimeric reads.

First, we applied GISTIC to the released copy number segments dataset, which verified the recurrent amplification of the *SOX2* locus (Figure S6A). The identified WGS GISTIC peak is larger in size as compared to the SNP-array-based GISTIC peak, which is likely due to the lower number of samples profiled by WGS and their modest sequencing coverage (ICGC/TCGA Pan-Cancer Analysis of Whole Genomes Consortium, 2020). Consistent with the SNP array data, the WGS GISTIC peak contains both *SOX2* and e1-e8 that we identified in squamous cancers (Figure S6A).

Next, as suggested by the reviewer, we analyzed the WGS structural variants dataset. Focusing on events that intersected or spanned the region between *SOX2* and the enhancer region, we found that 16 (14%) of the tumor samples have a duplication within or spanning this genomic window (Figure S6B). In addition, 11 (10%) of the tumors have duplications that harbor both the *SOX2* gene and the e1-e8 region and one tumor (SA53441 in Figure S6B) has a duplication that harbors *SOX2* and only e1-e2. Interestingly, we also observed recurrent duplications of just e1-e8 (without the *SOX2* gene) in four tumors samples (4%), although in one case the enhancer duplication is nested among larger duplications that contain both *SOX2* and the enhancer region. We then checked the SNP-array-based copy number data, which identified six samples harboring amplifications of only the enhancer alone (Figure S6B). The observation is reminiscent of our previous findings regarding duplications of enhancers adjacent to oncogenes such as *MYC* and *KLF5* (Zhang et al., 2016, 2018).

Figure S6 A. GISTIC result from squamous cancer WGS data. **B.** Upper: Structural variants identified by WGS analysis at the *SOX2*-e1 locus in squamous cancers. Bottom: SNP-array-based copy number data showing squamous cancer samples that have amplifications of the enhancer region alone.

These results further emphasize the importance of the e1 enhancer in squamous cancers. The data are now included in Figure S6 and described in page 7, paragraph 3: “Given the predominant role of the e1 enhancer in *SOX2* regulation, we sought to examine structural variants targeting e1 in squamous cancers. We downloaded whole-genome sequencing (WGS) data for 113 squamous cancers from the Pan-Cancer Atlas of Whole Genome (PCAWG) dataset. GISTIC analysis of the segment data validated the focal amplification of the *SOX2*-e1 locus (Figure S6A). We identified 16 tumor samples with tandem duplications at the *SOX2*-e1 region (Figure S6B). Duplications in 12 of the cases contain both *SOX2* and e1. Interestingly, four tumor samples harbor duplications of only the enhancer region without the *SOX2* gene (Figure S6B), reminiscent of our previous findings regarding duplications of *MYC* and *KLF5* enhancers. The presence of tandem duplications of just the enhancer region further highlights the importance of the e1 enhancer in squamous cancer.”

-Since *Sox2* is overlapped with the intron region of the *SOX2-OT* gene, the location of *SOX2-OT* should be indicated in the Figures, e.g. Figure 1 and 2. Supplementary Figure 2.

We have now added the location of *SOX2-OT* in the Figures 1, 2 S4 (previously S2), and S6.

-The correlation of *SOX2-OT* expression and amplification in the squamous cancers and glioma should also be examined.

Based on the reviewer’s suggestion, we have now performed correlation analyses for expression of *SOX2* and *SOX2-OT* versus *SOX2* amplifications in squamous cancers and gliomas.

In squamous cancers, we found that expression of both *SOX2* and *SOX2-OT* is positively correlated with *SOX2* amplifications (Figure R1 – for reviewer only). In addition, we found that samples with *SOX2* focal amplifications are associated with stronger expression of both *SOX2* and *SOX2-OT*, as compared to those with non-focal amplifications or those without amplifications (Figure S2).

This leads to an important question of whether or not *SOX2-OT* is a direct target of the e1 enhancer that is the focus of our study. Our new experiments showed that, although repression of e1 reduced *SOX2-OT* expression, ectopic re-expression of *SOX2* rescued the expression reduction (Figure S5C). In addition, *SOX2* binds at and near the promoter region of *SOX2-OT* (Figure S5D). Therefore, *SOX2-OT* is a target of the *SOX2* transcription factor, rather than a direct target of the e1 enhancer.

In gliomas, although *SOX2* expression is poorly correlated with *SOX2*

Figure R1 (for reviewer only): Correlation of *SOX2* amplifications versus *SOX2* and *SOX2-OT* expression in TCGA squamous cancers (upper) and gliomas (bottom).

Figure S2: Expression levels of *SOX2* and *SOX2-OT* in TCGA squamous cancers (upper) and gliomas (bottom) with *SOX2* focal amplifications, with *SOX2* non-focal amplifications, or without *SOX2* amplifications. P values are derived from t-tests.

Figure S5 C. Expression change (%) of the endogenous *SOX2* and *SOX2-OT* after e1 repression in KYSE140 cells with and without ectopic expression of *SOX2*. Primers targeting the *SOX2* 3’UTR region were used to distinguish the endogenous and ectopic *SOX2*. P value is derived from t-test: ***<math>< 0.001</math>. **D.** *SOX2* ChIP-seq profile at the *SOX2-OT* (P3 promoter isoform) locus.

is significantly higher in samples with *SOX2* focal amplifications as

compared to those with non-focal amplifications or those without amplifications. In contrast, *SOX2-OT* expression is not significantly different between gliomas with and without the focal amplifications (Figure S2). These results together suggest that the focal amplifications may contribute to *SOX2* but not *SOX2-OT* overexpression in gliomas.

These new results are now included in **Figure S2** and described in **page 4, paragraph 2**: “TCGA squamous cancers and gliomas with *SOX2* focal amplifications are associated with higher *SOX2* expression, as compared to samples with non-focal amplifications or samples without amplifications (Figure S2). *SOX2* overlaps with the *SOX2-OT* noncoding gene (Figure 1B). We found that *SOX2*-focally amplified squamous cancers are also associated with higher *SOX2-OT* expression, which was not observed in gliomas (Figure S2).”

-The location of all potential TF binding sites of *SOX2* and that included in the CRISPR cutting on the promoter and enhancer (e1 to e8) should be indicated in the figures.

To clarify, in the original manuscript we only performed CRISPR cutting of two *SOX* motifs at the e1 enhancer and found that the 2nd motif has clear effect on *SOX2* expression (Figure 4E). Based on the reviewer’s suggestion, we have now expanded the assay to *SOX* motifs in e2-e8 regions and the *SOX2* promoter (5kb +/- of *SOX2* TSS). In total, eight additional motifs (all annotated in Figure S8C now) can be cut by CRISPR/Cas9 (predicated based on relative positions of the motifs and their nearby PAM sequences). In the new experiment, we also included the 2nd *SOX* motif in e1 as our positive control. We showed that cutting the additional motifs had minimal or modest effects on *SOX2* expression. The results further highlight the importance of *SOX2* binding at the e1 enhancer in regulating *SOX2* expression. The data is now included in **Figure S8C**, and described in **page 9, paragraph 1**: “In addition, we also tested several additional *SOX* motifs in e2-e8 and the *SOX2* promoter, which showed that they have modest or minimal effect on *SOX2* expression (Figure S8C).”

Figure S8

Figure S8 C: Expression of *SOX2* in KYSE140 cells with cutting of *SOX* motifs in e2-e8 and *SOX2* promoter. The expression levels are normalized to the AAVS1 negative control. The 2nd *SOX* motif in e1 serves as a positive control for the experiment. P values are derived from t-tests: * <0.05 ; ** <0.01 ; *** <0.001 ; **** <0.0001 .

References

- Cho, S.W., Xu, J., Sun, R., Mumbach, M.R., Carter, A.C., Chen, Y.G., Yost, K.E., Kim, J., He, J., Nevins, S.A., et al. (2018). Promoter of lncRNA Gene PVT1 Is a Tumor-Suppressor DNA Boundary Element. *Cell* *173*, 1398-1412.e22.
- ICGC/TCGA Pan-Cancer Analysis of Whole Genomes Consortium (2020). Pan-cancer analysis of whole genomes. *Nature* *578*, 82–93.
- Zhang, X., Choi, P.S., Francis, J.M., Imielinski, M., Watanabe, H., Cherniack, A.D., and Meyerson, M. (2016). Identification of focally amplified lineage-specific super-enhancers in human epithelial cancers. *Nat. Genet.* *48*, 176–182.
- Zhang, X., Choi, P.S., Francis, J.M., Gao, G.F., Campbell, J.D., Ramachandran, A., Mitsuishi, Y., Ha, G., Shih, J., Vazquez, F., et al. (2018). Somatic Superenhancer Duplications and Hotspot Mutations Lead to Oncogenic Activation of the KLF5 Transcription Factor. *Cancer Discov.* *8*, 108–125.

Reviewers' Comments:

Reviewer #1:

Remarks to the Author:

The revised version answered my concerns and I recommend it will be accepted for publication.

Reviewer #2:

Remarks to the Author:

The new findings are interesting. All my questions and comments were addressed.

REVIEWERS' COMMENTS

Reviewer #1 (Remarks to the Author):

The revised version answered my concerns and I recommend it will be accepted for publication.

Reviewer #2 (Remarks to the Author):

The new findings are interesting. All my questions and comments were addressed.

RESPONSE:

We thank the reviewers for their positive comments on our work. Their previous comments have greatly helped us improve our manuscript.